# FAST AND SAMPLE-EFFICIENT DOMAIN ADAPTATION FOR AUTOENCODER-BASED E2E COMMUNICATION

## ABSTRACT

The problem of end-to-end learning of a communication system using an autoencoder has recently been shown to be a promising approach. We focus on the problem of *test-time domain adaptation* for such an autoencoder system whose channel is generatively-modeled using a mixture density network (MDN). Different from the setting of conventional training-time (unsupervised or semi-supervised) domain adaptation, here we have a fully-trained channel model and autoencoder from a source domain, that we would like to adapt to a target domain using only a small labeled dataset (and no unlabeled data). Moreover, since the distribution of the channel is expected to change frequently (e.g., a wireless link), the error rate of the autoencoder can degrade quickly, making it challenging to collect sufficient data for frequent retraining of the autoencoder. To address this, we propose a fast and sample-efficient method for adapting the autoencoder without modifying the encoder and decoder neural networks, and adapting *only the MDN* channel model. The method utilizes feature transformations at the decoder to compensate for changes in the channel distribution, and effectively present to the decoder samples close to the source distribution. Experimental evaluation on simulated datasets and real mmWave wireless channels demonstrate that the proposed method can adapt the MDN channel using very limited number of samples, and improve or maintain the error rate of the autoencoder under changing channel conditions.

## 1 INTRODUCTION

End-to-end (e2e) learning of communication systems using autoencoders has been recently shown to be a promising approach for designing the next generation of wireless networks (O'Shea & Hoydis, 2017; Dörner et al., 2018; Aoudia & Hoydis, 2019; O'Shea et al., 2019; Ye et al., 2018; Wang et al., 2017). This new paradigm is a viable alternative to optimize communication for diverse applications, hardware, and environments (Hoydis et al., 2021). It is particularly promising for dense deployments of low-cost transceivers where there is interference between devices and hardware imperfections that are difficult to model. The key idea of e2e learning for a communication system is to use an autoencoder architecture to model and learn the transmitter and receiver jointly using neural networks in order to minimize an e2e performance metric such as the block error rate (BLER) (O'Shea & Hoydis, 2017). The channel (the propagation medium and transceiver imperfections) can be represented as a stochastic transfer function that transforms its input $\mathbf{x} \in \mathbb{R}^d$ to an output $\mathbf{y} \in \mathbb{R}^d$. It can be regarded as a black-box that is non-linear and non-differentiable due to hardware imperfections (*e.g.*, quantization and amplifiers). Since autoencoders are trained using stochastic gradient descent (SGD)-based optimization (O'Shea & Hoydis, 2017), it is challenging to work with a black-box channel that is not differentiable. One approach to address this problem is by using a known mathematical model of the channel (*e.g.*, additive Gaussian noise). Use of such models enables the computation of gradients of the loss function with respect to the autoencoder parameters via backpropagation. However, such standard channel models do not capture well the realistic channel effects, as shown in (Aoudia & Hoydis, 2018). Alternatively, recent works have proposed to learn the channel using deep generative models that approximate $p(\mathbf{y} \mid \mathbf{x})$, the conditional probability density of the channel output $\mathbf{y}$ given the channel input $\mathbf{x}$, using generative adversarial networks (GANs) (O'Shea et al., 2019; Ye et al., 2018), mixture density networks (MDNs) (García Martí et al., 2020), and conditional variational autoencoders (VAEs) (Xia et al., 2020). The use of a differentiable generative model of the channel enables SGD-based training of the autoencoder, while also capturing realistic channel effects better than standard models.

Although this e2e optimization with real channels learned from data can improve the physical layer design for communication systems, in reality, channels often change, requiring collection of a large number of samples and retraining the channel model and autoencoder frequently. For this reason, *adapting the learned conditional probability density of the channel as often as possible using only a small number of samples* is required for good communication performance. Prior works have (to be best of our knowledge) not addressed the adaptation problem for autoencoder-based e2e learning, which is crucial for the real-time deployment of such a system under frequently-changing channel conditions. In this paper, we study the problem of domain adaptation (DA) of autoencoders using an MDN as the channel model. In contrast to the conventional DA setting, where one has access to a large unlabeled dataset and none or a small labeled dataset from the target domain (Jiang, 2008; Ben-David et al., 2006), here we consider DA where we only have access to a small labeled dataset from the target domain. This setting applies to our problem since the channel distribution changes frequently, and we only get to collect a small number of samples at a time from the target domain.

We make the following **contributions**: 1) We propose a fast and sample-efficient method for adapting a generative MDN (used for modeling the channel) based on the properties of Gaussian mixtures. 2) Based on the MDN adaptation, we propose efficient input-transformation methods at the decoder that compensate for changes in the class-conditional channel distribution, and decrease or maintain the error rate of the autoencoder without requiring any retraining of the encoder and decoder networks. [1]

## 2 RELATED WORK

**Mixture Density Networks.** MDNs were first introduced by (Bishop, 1994), providing a new framework for modeling complex conditional densities using neural networks. Recently, (García Martí et al., 2020) proposed to use an MDN to learn the wireless channel since Gaussian mixtures, with their strong approximation capability (Kostantinos, 2000), can accurately capture the channel distribution given sufficient parametric complexity and data. García Martí et al. (2020) also proposed to adapt the MDN model to changing channel conditions by fine-tuning the MDN with a small set of samples from a new distribution. To the best of our knowledge, (Li et al., 2020) is the only other work to study the problem of adapting an MDN. They address the speaker identification problem, and propose a gradient-based meta-learning algorithm for MDN that learns to transfer knowledge from an existing set of speakers to a new speaker using a small number of labeled samples.

**Domain Adaptation, Transfer Learning, and Few-Shot Learning.** Recent approaches for DA such as DANN (Ganin et al., 2016) based on adversarial learning of a shared representation between the source and target domains (Ganin & Lempitsky, 2015; Ganin et al., 2016; Long et al., 2018; Saito et al., 2018; Zhao et al., 2019; Johansson et al., 2019) have achieved much success on computer vision and natural language processing tasks. Their high-level idea is to adversarially learn a shared feature representation for which inputs from the source and target distributions are nearly indistinguishable to a *domain discriminator* DNN, such that a *label predictor* DNN using this representation and trained using labeled data from only the source domain also generalizes well to the target domain. Adversarial DA methods are not suitable for our problem because of the high imbalance in the number of source and target domain samples (hard to learn a good domain discriminator). Also, adversarial DA methods being heavy on computational and data requirement, are not well-suited for fast and frequent test-time DA. Related frameworks such as transfer learning (Long et al., 2015; 2016), model-agnostic meta-learning (Finn et al., 2017), domain-adaptive few-shot learning (Zhao et al., 2021; Sun et al., 2019), and supervised DA (Motiian et al., 2017a;b) also deal with the problem of frequent adaptation based on a small number of samples. Most of them are not directly applicable to our problem because they primarily address novel classes (with potentially different distributions) and knowledge transfer from existing to novel tasks. Motiian et al. (2017a) is closely related to our work since they also deal with a target domain that only has a small labeled dataset and has the same set of classes (label space). The *key difference* is that (Motiian et al., 2017a) address the training-time supervised DA problem, while we focus on the test-time supervised DA problem. In the test-time setting, frequent and fast adaptation of a trained source-domain classifier (here the decoder) is required. It can be computationally challenging to adopt the adversarial DA method of Motiian et al. (2017a) that would have to be retrained on every new batch of target domain samples.

---

[1] Code base for our work: `anonymous.4open.science/r/domain_adaptation-7C0D/`

## 3 FAST ADAPTATION OF THE MDN CHANNEL MODEL

**Notations and Problem Setup.** We denote vectors and matrices by boldface symbols. We define $\mathbb{1}(c)$ to be an indicator function that takes the value 1 (0) when predicate $c$ is true (false). For any integer $n > 1$, we define $[n] = \{1, \cdots, n\}$. We denote the one-hot-coded vector of all zeros except a 1 at position $i \in [m]$ by $\mathbf{1}_i \in \{0, 1\}^m$. The probability density function (pdf) of a multivariate Gaussian with mean vector $\boldsymbol{\mu}$ and covariance matrix $\boldsymbol{\Sigma}$ is denoted by $\mathcal{N}(\cdot \,|\, \boldsymbol{\mu}, \boldsymbol{\Sigma})$. We consider a single-input, single-output autoencoder-based communication system as shown in Fig. 7 in Appendix A. The transmitter or encoder part of the autoencoder is as a multi-layer, feed-forward neural network (NN) that takes as input the one-hot-coded representation $\mathbf{1}_s$ of a message $s \in \mathcal{S} := \{1, 2, \cdots, m\}$, and produces an encoded symbol vector $\mathbf{x} = \mathbf{E}_{\boldsymbol{\theta}_e}(\mathbf{1}_s) \in \mathbb{R}^d$. The receiver or decoder part is also a multi-layer, feed-forward NN that takes the channel output $\mathbf{y} \in \mathbb{R}^d$ as its input and predicts a probability distribution over the $m$ messages. The input-output mapping of the decoder NN can be defined as $\mathbf{D}_{\boldsymbol{\theta}_d}(\mathbf{y}) := [P_{\boldsymbol{\theta}_d}(1 \,|\, \mathbf{y}), \cdots, P_{\boldsymbol{\theta}_d}(m \,|\, \mathbf{y})]^T$. The message $s$ is equivalent to a class label and the encoded symbol vector $\mathbf{x} = \mathbf{E}_{\boldsymbol{\theta}_e}(\mathbf{1}_s)$ is like a representation vector for label $s$. Table 3 in the Appendix provides a quick reference for the notations used in the paper.

**Channel Modeling using MDN.** In this work, we use an MDN with Gaussian components to learn the channel conditional density $P(\mathbf{y} \,|\, \mathbf{x})$. MDNs can model complex conditional densities by combining a feed-forward neural network with a standard parametric mixture model (Bishop, 1994). The MDN learns to predict the parameters of the mixture model $\boldsymbol{\phi}(\mathbf{x})$ as a function of the channel input $\mathbf{x}$. This can be expressed as $\boldsymbol{\phi}(\mathbf{x}) = \mathbf{M}_{\boldsymbol{\theta}_c}(\mathbf{x})$, where $\boldsymbol{\theta}_c$ is the parameter vector of the neural network. The parameters of the mixture model defined by the MDN are a concatenation of the parameters from the $k$ density components, *i.e.*, $\boldsymbol{\phi}(\mathbf{x})^T = [\boldsymbol{\phi}_1(\mathbf{x})^T, \cdots, \boldsymbol{\phi}_k(\mathbf{x})^T]$. A more detailed background on MDNs and autoencoder-based end-to-end learning is given in Appendix A.

Consider the setting where the channel state (and therefore its conditional distribution) is changing over time due to *e.g.*, environmental factors. Let $P(\mathbf{y} \,|\, \mathbf{x})$ denote the (unknown) *source channel distribution* underlying the dataset $\mathcal{D}_c$ used to train the MDN $\mathbf{M}_{\boldsymbol{\theta}_c}(\mathbf{x})$. With a sufficiently large dataset and a suitable choice of $k$, the Gaussian mixture learned by the MDN $P_{\boldsymbol{\theta}_c}(\mathbf{y} \,|\, \mathbf{x})$ can closely approximate $P(\mathbf{y} \,|\, \mathbf{x})$.

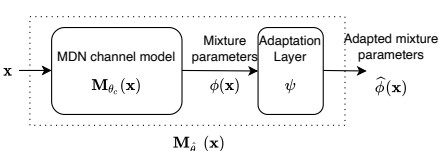

Figure 1: Proposed MDN adaptation overview.

Let $\mathcal{D}_c^{(t)} = \{(\mathbf{x}_n^{(t)}, \mathbf{y}_n^{(t)}), \ n = 1, \cdots, N_c^{(t)}\}$ denote a small set of iid samples ($N_c^{(t)} \ll |\mathcal{D}_c|$) from an unknown *target channel distribution* $P^{(t)}(\mathbf{y} \,|\, \mathbf{x})$, which is potentially different from $P(\mathbf{y} \,|\, \mathbf{x})$ but not by a large deviation. Our goal is to adapt the MDN (and therefore the underlying mixture density) using $\mathcal{D}_c^{(t)}$ such that it closely approximates $P^{(t)}(\mathbf{y} \,|\, \mathbf{x})$. Note that the space of inputs to the MDN is the finite set of modulated symbols $\mathcal{X} = \{\mathbf{E}_{\boldsymbol{\theta}_e}(\mathbf{1}_s), \ s \in \mathcal{S}\}$ (referred to as a constellation), with each symbol corresponding to a unique message $s \in \mathcal{S}$.

### 3.1 TRANSFORMATION BETWEEN GAUSSIAN MIXTURES

$$P_{\boldsymbol{\theta}_c}(\mathbf{y} \,|\, \mathbf{x}) = \sum_{i=1}^{k} \pi_i(\mathbf{x}) \, N\left(\mathbf{y} \,|\, \boldsymbol{\mu}_i(\mathbf{x}), \boldsymbol{\sigma}_i^2(\mathbf{x})\right) \quad (1)$$

$$P_{\widehat{\boldsymbol{\theta}}_c}(\mathbf{y} \,|\, \mathbf{x}) = \sum_{i=1}^{k} \widehat{\pi}_i(\mathbf{x}) \, N\left(\mathbf{y} \,|\, \widehat{\boldsymbol{\mu}}_i(\mathbf{x}), \widehat{\boldsymbol{\sigma}}_i^2(\mathbf{x})\right). \quad (2)$$

Consider the Gaussian mixtures corresponding to the source and target channel conditional densities, where $\boldsymbol{\theta}_c$ and $\widehat{\boldsymbol{\theta}}_c$ are the parameter vectors of the original and adapted MDN. Here $\boldsymbol{\mu}_i(\mathbf{x}) \in \mathbb{R}^d$ is the mean vector, $\boldsymbol{\sigma}_i^2(\mathbf{x}) \in \mathbb{R}_+^d$ is the variance vector, and $\pi_i(\mathbf{x}) \in [0, 1]$ is the prior probability (weight) of component $i$ for the original mixture. We have assumed that the Gaussian components have a diagonal covariance matrix, with $\boldsymbol{\sigma}_i^2(\mathbf{x})$ being the diagonal elements [2]. The mixture weights are parameterized using the softmax function as $\pi_i(\mathbf{x}) = e^{\alpha_i(\mathbf{x})} / \sum_{j=1}^{k} e^{\alpha_j(\mathbf{x})}, \ \forall i$. The MDN simply predicts the un-normalized weights $\alpha_i(\mathbf{x}) \in \mathbb{R}$ or the prior logits. The parameter vector of component $i$ is defined as $\boldsymbol{\phi}_i(\mathbf{x})^T = [\alpha_i(\mathbf{x}), \boldsymbol{\mu}_i(\mathbf{x})^T, \boldsymbol{\sigma}_i^2(\mathbf{x})^T]$, and the MDN output $\boldsymbol{\phi}(\mathbf{x})$ has dimension $k\,(2d + 1)$. The adapted MDN predicts the parameters of the target Gaussian mixture $\widehat{\boldsymbol{\phi}}(\mathbf{x}) = \mathbf{M}_{\widehat{\boldsymbol{\theta}}_c}(\mathbf{x})$ as shown in Fig. 1. The parameters of the adapted MDN and Gaussian mixture are similarly defined, with

---

[2]The diagonal covariance assumption does *not* imply conditional independence of $\mathbf{y}$ as long as $k > 1$.

a hat in the notation. We next summarize the feature and parameter transformations required for mapping the component densities of one Gaussian mixture to another. As shown in Appendix C.1, the transformation between any two multivariate Gaussians $\mathbf{y} \sim N(\cdot \,|\, \boldsymbol{\mu}, \boldsymbol{\Sigma})$ and $\widehat{\mathbf{y}} \sim N(\cdot \,|\, \widehat{\boldsymbol{\mu}}, \widehat{\boldsymbol{\Sigma}})$ can be achieved by the transformation: $\widehat{\mathbf{y}} = \mathbf{C}\,(\mathbf{y} - \boldsymbol{\mu}) + \mathbf{A}\,\boldsymbol{\mu} + \mathbf{b}$, where the mean vector and covariance matrix of the two Gaussians are related as follows: $\widehat{\boldsymbol{\mu}} = \mathbf{A}\,\boldsymbol{\mu} + \mathbf{b}$ and $\widehat{\boldsymbol{\Sigma}} = \mathbf{C}\,\boldsymbol{\Sigma}\,\mathbf{C}^T$.

**Affine and Inverse-Affine Feature Transformations.** Applying the above result to our MDN with $k$ components, we define the affine feature transformation for a given symbol $\mathbf{x}$ and component $i$, mapping from $\mathbf{y} \sim N(\cdot \,|\, \boldsymbol{\mu}_i(\mathbf{x}), \boldsymbol{\sigma}_i^2(\mathbf{x}))$ to $\widehat{\mathbf{y}} \sim N(\cdot \,|\, \widehat{\boldsymbol{\mu}}_i(\mathbf{x}), \widehat{\boldsymbol{\sigma}}_i^2(\mathbf{x}))$ as

$$\widehat{\mathbf{y}} = \mathbf{g}_{\mathbf{x}i}(\mathbf{y}) := \mathbf{C}_i\,(\mathbf{y} - \boldsymbol{\mu}_i(\mathbf{x})) + \mathbf{A}_i\,\boldsymbol{\mu}_i(\mathbf{x}) + \mathbf{b}_i, \ \ \mathbf{x} \in \mathcal{X}, \ i \in [k]. \tag{3}$$

It is straightforward to also define the inverse-affine transformation from $\widehat{\mathbf{y}}$ to $\mathbf{y}$ as

$$\mathbf{y} = \mathbf{g}_{\mathbf{x}i}^{-1}(\widehat{\mathbf{y}}) := \mathbf{C}_i^{-1}\,(\widehat{\mathbf{y}} - \mathbf{A}_i\,\boldsymbol{\mu}_i(\mathbf{x}) - \mathbf{b}_i) + \boldsymbol{\mu}_i(\mathbf{x}), \ \ \mathbf{x} \in \mathcal{X}, \ i \in [k]. \tag{4}$$

For the case of diagonal covariances, we constrain $\mathbf{C}_i$ to be diagonal. These feature transformations will be used for aligning the target and source class-conditional distributions of the decoder input.

**Parameter Transformations.** The corresponding transformations between the source and target Gaussian mixture parameters for any symbol $\mathbf{x} \in \mathcal{X}$ and component $i \in [k]$ are given by

$$\widehat{\boldsymbol{\mu}}_i(\mathbf{x}) = \mathbf{A}_i\,\boldsymbol{\mu}_i(\mathbf{x}) + \mathbf{b}_i, \ \ \widehat{\boldsymbol{\sigma}}_i^2(\mathbf{x}) = \mathbf{C}_i^2\,\boldsymbol{\sigma}_i^2(\mathbf{x}), \ \text{and} \ \ \widehat{\alpha}_i(\mathbf{x}) = \beta_i\,\alpha_i(\mathbf{x}) + \gamma_i, \tag{5}$$

where $\mathbf{A}_i \in \mathbb{R}^{d \times d}$ and $\mathbf{b}_i \in \mathbb{R}^d$ transform the means; $\mathbf{C}_i = \mathrm{diag}(c_{i1}, \cdots, c_{id})$ is a diagonal scale matrix for the variances; and $\beta_i \in \mathbb{R}$ and $\gamma_i \in \mathbb{R}$ are the scale and offset for the prior logits. The vector of all *adaptation parameters* to be optimized is defined as $\boldsymbol{\psi}^T = [\boldsymbol{\psi}_1^T, \cdots, \boldsymbol{\psi}_k^T]$, where $\boldsymbol{\psi}_i^T$ contains the affine-transformation parameters from component $i$. The number of adaptation parameters (dimension of $\boldsymbol{\psi}$) is given by $k\,(d^2 + 2\,d + 2)$. This is typically much smaller than the number of MDN parameters (weights and biases from all layers), even for shallow fully-connected NNs. In Fig. 1, the adaptation layer mapping $\phi(\mathbf{x})$ to $\widehat{\phi}(\mathbf{x})$ basically implements the parameter transformations defined in Eq. (5).

**Assumptions and Key Insight.** The proposed adaptation of the MDN is based on the affine-transformation property of multivariate Gaussians, *i.e.*, one can transform between any two multivariate Gaussians through an affine transformation.
*Assumption 1: The source and target Gaussian mixtures have the same number of components.*
This is a practical assumption we make in order to not have to change the architecture of the MDN. Adding or removing components would require a change to the output layer of the MDN. Also, this assumption can be practically justified when $k$ is chosen to be sufficiently large.
*Assumption 2: The two mixtures have a one-to-one correspondence between the components.*
This assumption makes in mathematically convenient to derive a closed-form expression for the KL-divergence between two Gaussian mixtures, which would not be possible in the general case.

Based on the above assumptions, we can formulate the MDN adaptation as an equivalent problem of finding the optimal set of affine transformations (one per-component) from the source to the target Gaussian mixture. This is a *much smaller* problem compared to optimizing the weights of all the MDN layers. Moreover, the affine transformations are bijective, allowing the feature and parameter mapping to be applied in the inverse direction. To reduce the possibility of the adapted MDN finding bad solutions due to the small-sample regime, we include a regularization term based on the KL-divergence (KLD) in the adaptation objective.

## 3.2 DIVERGENCE BETWEEN THE SOURCE AND TARGET DISTRIBUTIONS

For the pair of Gaussian mixtures (Eqs. (1) and (2)), based on Assumption 2 we can derive a closed-form expression for the Kullback-Leibler divergence between them, given by

$$\overline{D}(P_{\boldsymbol{\theta}_c}, P_{\widehat{\boldsymbol{\theta}}_c}) = \mathbb{E}_{P_{\boldsymbol{\theta}_c}}\left[\log \frac{P_{\boldsymbol{\theta}_c}(\mathbf{y}, K \,|\, \mathbf{x})}{P_{\widehat{\boldsymbol{\theta}}_c}(\mathbf{y}, K \,|\, \mathbf{x})}\right] = \sum_{\mathbf{x} \in \mathcal{X}} p(\mathbf{x}) \sum_{i=1}^{k} \pi_i(\mathbf{x}) \log \frac{\pi_i(\mathbf{x})}{\widehat{\pi}_i(\mathbf{x})}$$

$$+ \sum_{\mathbf{x} \in \mathcal{X}} p(\mathbf{x}) \sum_{i=1}^{k} \pi_i(\mathbf{x}) \, D_{\mathrm{KL}}\left(N\left(\cdot \,|\, \boldsymbol{\mu}_i(\mathbf{x}), \boldsymbol{\sigma}_i^2(\mathbf{x})\right), N\left(\cdot \,|\, \widehat{\boldsymbol{\mu}}_i(\mathbf{x}), \widehat{\boldsymbol{\sigma}}_i^2(\mathbf{x})\right)\right). \tag{6}$$

A detailed derivation of this result and the final expression for the KLD as a function of $\psi$ are given in Appendix C.2. The first term in Eq. (6) is the KLD between the component prior probabilities, which simplifies into a function of the parameters $[\beta_1, \gamma_1, \cdots, \beta_k, \gamma_k]$. The second term in Eq. (6) involves the KLD between two multivariate Gaussians (a standard result), which also simplifies into a function of $\psi$. To make the dependence on $\psi$ explicit, the KLD is henceforth denoted by $\overline{D}_\psi(P_{\boldsymbol{\theta}_c}, P_{\widehat{\boldsymbol{\theta}}_c})$.

### 3.3 MDN Adaptation Objectives

We consider two scenarios for adaptation: 1) Generative adaptation of the MDN in isolation and 2) Discriminative adaptation of the MDN as the channel model for the autoencoder. In the first case, the goal of adaptation is to find a good generative model for the target data distribution, while in the second case, the goal is to improve the classification performance of the autoencoder on the target data distribution. In both cases, we formulate the MDN adaptation as a minimization problem with a regularized negative log-likelihood objective, where the regularization term penalizes solutions with a large KLD between the source and target Gaussian mixtures.

**Generative Adaptation.**  The data-dependent term in the adaptation objective is the regularized negative conditional log-likelihood (CLL) of the target dataset:

$$J_{\text{CLL}}(\boldsymbol{\psi}\,;\lambda) \;=\; \frac{-1}{N_c^{(t)}} \sum_{n=1}^{N_c^{(t)}} \log P_{\widehat{\boldsymbol{\theta}}_c}(\mathbf{y}_n^{(t)} \,|\, \mathbf{x}_n^{(t)}) \;+\; \lambda\,\overline{D}_\psi(P_{\boldsymbol{\theta}_c}, P_{\widehat{\boldsymbol{\theta}}_c}), \tag{7}$$

where $\widehat{\boldsymbol{\mu}}_i(\mathbf{x}), \widehat{\boldsymbol{\sigma}}_i^2(\mathbf{x})$ and $\widehat{\alpha}_i(\mathbf{x})$ as a function of $\psi$ are given by Eq. (5). The parameters of the original mixture density $\alpha_i(\mathbf{x}), \boldsymbol{\mu}_i(\mathbf{x}), \boldsymbol{\sigma}_i^2(\mathbf{x}), \forall i$ are constants since they have no dependence on $\psi$. The regularization constant $\lambda \geq 0$ controls the allowed KLD between the source and target Gaussian mixtures. Small values of $\lambda$ weight the CLL term more, allowing more exploration in the adaptation; large values of $\lambda$ impose a strong regularization to constrain the space of target distributions.

**Discriminative Adaptation.**  With the goal of improving the accuracy of the decoder in recovering the transmitted symbol $\mathbf{x}$ from $\mathbf{y}$, the data-dependent term in the adaptation objective (7) is replaced with the symbol posterior log-likelihood (PLL) as follows:

$$J_{\text{PLL}}(\boldsymbol{\psi}\,;\lambda) \;=\; \frac{-1}{N_c^{(t)}} \sum_{n=1}^{N_c^{(t)}} \log P_{\widehat{\boldsymbol{\theta}}_c}(\mathbf{x}_n^{(t)} \,|\, \mathbf{y}_n^{(t)}) \;+\; \lambda\,\overline{D}_\psi(P_{\boldsymbol{\theta}_c}, P_{\widehat{\boldsymbol{\theta}}_c}). \tag{8}$$

The symbol posterior $P_{\widehat{\boldsymbol{\theta}}_c}(\mathbf{x}\,|\,\mathbf{y})$ is computed using Bayes rule, and the symbol prior $\{p(\mathbf{x} = \mathbf{E}_{\boldsymbol{\theta}_e}(\mathbf{1}_s)),\;\; s = 1, \cdots, m\}$ is estimated from the source domain training set.

We observe that the adaptation objectives are smooth and nonconvex function of $\psi$. Also, computation of the objective and its gradient w.r.t $\psi$ are inexpensive operations since **i)** they *do not require forward and back-propagation steps* through the layers of the MDN and **ii)** both $N_c^{(t)}$ and the dimension of $\psi$ are relatively small. Therefore, we use the BFGS quasi-newton method (Nocedal & Wright, 2006) for minimization, instead of SGD-based large-scale learning methods (*e.g.*, Adam). The regularization constant $\lambda$ is a hyper-parameter of the proposed method, and we propose to set its value automatically using a validation metric based on the inverse-affine transformation from the target to the source distribution (see Appendix C.3).

## 4 Adaptation of Autoencoder-Based Communication System

In this section, we discuss how the proposed MDN adaptation can be combined with an autoencoder-based communication system to adapt the decoder to changes in the channel conditions. Recall that the decoder is basically a classifier that predicts the most-probable input message from the received channel output $\mathbf{y}$. When the decoder operates in a new (target) channel environment, different from the one it was trained on, its classification accuracy can degrade due to the distribution change. Specif-

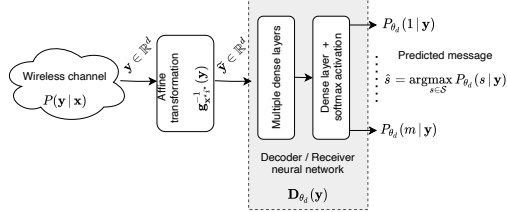

Figure 2: Adapted decoder with affine transformations.

ically, any change in the channel conditions reflects as changes in the class-conditional density of

the decoder's input, *i.e.*, $\{P(\mathbf{y}\,|\,s),\ s \in \mathcal{S}\}$ changes [3]. We propose to address this, by designing transformations to the decoder's input that can compensate for changes in the channel distribution, and effectively present transformed inputs that are close to the source distribution on which the decoder was trained. Our method does not require any change or adaptation to the trained encoder and decoder networks, making it fast and suitable for the small-sample setting. We next propose two such input transformation methods for the decoder.

## 4.1 Adapted Decoder Based on Affine Feature Transformations

Consider the same problem setup as § 3, where we observe a small dataset of samples from the target channel distribution. Suppose we have adapted the MDN channel by optimizing over the parameters $\psi$, we can use the inverse-affine feature transformations (defined in Eq. (4)) to transform the channel output $\mathbf{y}$ from a component of the target Gaussian mixture to the same component of the source Gaussian mixture. However, this transformation requires knowledge of both the channel input $\mathbf{x}$ and the mixture component $i$, which are not observed (latent) at the decoder. We address this by first determining the most-probable pair of channel input and mixture component for a given $\mathbf{y}$ (using the MAP rule), and applying the corresponding inverse-affine feature transformation as

$$\widetilde{y} = \mathbf{g}_{\mathbf{x}^\star i^\star}^{-1}(\mathbf{y}) \quad \text{where} \quad \mathbf{x}^\star, i^\star = \underset{\mathbf{x} \in \mathcal{X}, i \in [k]}{\arg\max} P_{\widehat{\boldsymbol{\theta}}_c}(\mathbf{x}, i \,|\, \mathbf{y}). \tag{9}$$

The joint posterior over the channel input $\mathbf{x}$ and mixture component $i$, given the channel output $\mathbf{y}$ is based on the adapted (target) Gaussian mixture, given by

$$P_{\widehat{\boldsymbol{\theta}}_c}(\mathbf{x}, i \,|\, \mathbf{y}) \;=\; \frac{p(\mathbf{x})\,\widehat{\pi}_i(\mathbf{x})\,N(\mathbf{y}\,|\,\widehat{\boldsymbol{\mu}}_i(\mathbf{x}), \widehat{\boldsymbol{\sigma}}_i^2(\mathbf{x}))}{\sum_{\mathbf{x}'}\sum_j p(\mathbf{x}')\,\widehat{\pi}_j(\mathbf{x}')\,N(\mathbf{y}\,|\,\widehat{\boldsymbol{\mu}}_j(\mathbf{x}'), \widehat{\boldsymbol{\sigma}}_j^2(\mathbf{x}'))}.$$

The adapted decoder based on the above affine feature transformation (see Fig. 2) is defined as

$$\widehat{\mathbf{D}}_{\boldsymbol{\theta}_d}(\mathbf{y}) \;:=\; \mathbf{D}_{\boldsymbol{\theta}_d}(\mathbf{g}_{\mathbf{x}^\star i^\star}^{-1}(\mathbf{y})) \;=\; \mathbf{D}_{\boldsymbol{\theta}_d} \circ \mathbf{g}_{\mathbf{x}^\star i^\star}^{-1}(\mathbf{y}). \tag{10}$$

## 4.2 Adapted Decoder Based on MAP Symbol Estimation

In the previous method, an input transformation layer is introduced at the decoder *only* during adaptation, but not during training of the autoencoder. Alternatively, we propose an input transformation layer at the decoder that takes the channel output $\mathbf{y}$ and produces a best estimate of the encoded symbol $\widehat{\mathbf{x}}$, which is then given as input to the decoder as shown in Fig. 3. This input transformation layer is included during the autoencoder training as a fixed non-linear transformation that does not have any trainable parameters. Since the decoder is trained to predict using $\widehat{\mathbf{x}}$ instead of $\mathbf{y}$, it is inherently *robust to changes* in the distribution of $\mathbf{y}$.

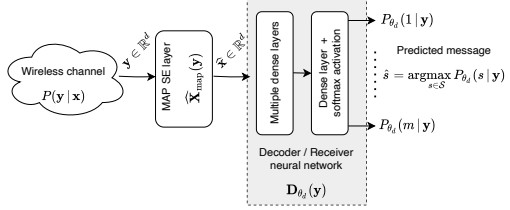

Figure 3: Adapted decoder with MAP SE.

Given a generative model of the channel conditional density using Gaussian mixtures, we can estimate the plug-in Bayes posterior distribution of $\mathbf{x}$ given $\mathbf{y}$, $P_{\boldsymbol{\theta}_c}(\mathbf{x}\,|\,\mathbf{y})$ (ref. Eq. (27)). From this, we can find the MAP estimate of $\mathbf{x}$ given $\mathbf{y}$ as

$$\widehat{\mathbf{X}}_{\text{map}}(\mathbf{y}) \;=\; \underset{\mathbf{x} \in \mathcal{X}}{\arg\max}\; P_{\boldsymbol{\theta}_c}(\mathbf{x}\,|\,\mathbf{y}) \;=\; \underset{\mathbf{x} \in \mathcal{X}}{\arg\max}\; \log P_{\boldsymbol{\theta}_c}(\mathbf{y}\,|\,\mathbf{x}) \,+\, \log p(\mathbf{x}). \tag{11}$$

The adapted decoder based on this input transformation, referred to as the *MAP symbol estimation (SE)* layer, is defined as

$$\widehat{\mathbf{D}}_{\boldsymbol{\theta}_d}(\mathbf{y}) \;:=\; \mathbf{D}_{\boldsymbol{\theta}_d}(\widehat{\mathbf{X}}_{\text{map}}(\mathbf{y})) \;=\; \mathbf{D}_{\boldsymbol{\theta}_d} \circ \widehat{\mathbf{X}}_{\text{map}}(\mathbf{y}), \tag{12}$$

and illustrated in Fig. 3. Whenever the MDN model is adapted to changes in the channel distribution, resulting in a new MDN with parameters $\widehat{\boldsymbol{\theta}}_c$, the MAP SE layer is also updated using $\widehat{\boldsymbol{\theta}}_c$. This input transformation shields the decoder from changes to the distribution of the channel output $\mathbf{y}$.

---

[3]For this generative model, it is easy to see that the class-conditional density is equal to the channel-conditional density, *i.e.*, $P(\mathbf{y}\,|\,s) = P(\mathbf{y}\,|\,\mathbf{E}_{\boldsymbol{\theta}_e}(\mathbf{1}_s))$, $\forall s$. Hence, by adapting the MDN, we are effectively also adapting the class-conditional density of the decoder's input.

Since the MAP SE layer is also included in the autoencoder during training, the non-differentiable `argmax` function presents an obstacle to training the autoencoder using backpropagation. We address this by using a temperature-scaled softmax approximation to the `argmax`, which is differentiable and provides a close approximation for small temperature values. This approximation is used only during training, whereas the exact `argmax` is used during inference. Details on this approximation, and a modified autoencoder training algorithm with temperature annealing are discussed in Appendix C.5.

**Comments.** The proposed input transformation methods at the decoder have some similarities to equalization methods used in communication receivers (Goldsmith, 2005). However, our problem setting considers a memoryless channel, and does not deal with intersymbol interference (ISI), which is the main focus of equalization methods. A key advantage of the proposed adaptation is that it is very computationally efficient to implement at the receiver of a communication system. A discussion of the computational complexity of the proposed methods is given in Appendix C.4.

## 5 EXPERIMENTAL EVALUATION

We implemented the MDN, communication autoencoder, and the adaptation methods in Python using TensorFlow (Abadi et al., 2015) and TensorFlow Probability. We used the following setting in our experiments. The size of the message set $m$ was fixed to 16, corresponding to 4 bits. The dimension of the encoding (output of the encoder) $d$ was set to 2, and the number of mixture components $k$ was set to 5. More details on the experimental setup, neural network architecture, and the hyper-parameters are given in Appendix D.1. The generative adaptation objective (7) is used for the experiments in § 5.1, where the MDN is adapted in isolation. The discriminative adaptation objective (8) is used for the experiments in § 5.2 and § 5.3, where the MDN is adapted as part of the autoencoder.

### 5.1 MDN ADAPTATION ON SIMULATED CHANNELS

We evaluate the proposed adaptation method for an MDN (§ 3) on simulated channel variations based on models commonly used for wireless communication. Specifically, we use the following channel models: i) additive white Gaussian noise (AWGN), ii) Ricean fading, and iii) Uniform or flat fading (Goldsmith, 2005). Details on these channel models and calculation of the their signal-to-noise ratio (SNR) are provided in Appendix E. In each case, the MDN is first trained on a large dataset simulated from a particular type of channel model (*e.g.*, AWGN), referred to as the source channel. The trained MDN is then adapted using a small dataset from a different type of channel model (*e.g.*, Ricean fading), referred to as the target channel. We used a standard constellation corresponding to quadrature amplitude modulation of 16 symbols, referred to as 16-QAM (Goldsmith, 2005), as inputs to the channel. A training set of 25000 samples from the source channel is used to train the MDN. The size of the adaptation dataset from the target channel is varied over a few different values – 5, 10, 15, and 20 samples per symbol, corresponding to target datasets of size 80, 160, 240, and 320.

Table 1: Log-likelihood of the MDN adaptation methods on simulated channel variations

| Source channel | Target channel | #Target samples | Proposed | | Transfer | | Transfer-last-layer | |
|---|---|---|---|---|---|---|---|---|
| | | | median | 95% CI | median | 95% CI | median | 95% CI |
| AWGN | Uniform fading | 80 | **-0.49** | (-3.89, 0.45) | -6.97 | (-17.92, -1.72) | -2.09 | (-6.50, -0.21) |
| | | 160 | **-0.43** | (-2.32, 0.48) | -1.65 | (-4.41, -0.14) | -0.79 | (-1.90, 0.13) |
| | | 240 | -0.58 | (-1.94, 0.52) | -0.74 | (-2.07, 0.25) | **-0.35** | (-1.32, 0.51) |
| | | 320 | -0.22 | (-2.27, 0.63) | -0.40 | (-1.35, 0.31) | **-0.19** | (-1.16, 0.54) |
| AWGN | Ricean fading | 80 | **1.17** | (0.68, 1.33) | -1.78 | (-6.88, 0.22) | -0.64 | (-6.86, 0.96) |
| | | 160 | **1.26** | (0.51, 1.39) | 0.37 | (-1.10, 0.88) | 0.55 | (-0.71, 1.22) |
| | | 240 | **1.31** | (-0.09, 1.39) | 0.91 | (0.28, 1.17) | 1.00 | (0.42, 1.29) |
| | | 320 | **1.27** | (0.70, 1.41) | 1.07 | (0.73, 1.22) | 1.14 | (0.86, 1.32) |
| Ricean fading | Uniform fading | 80 | **-0.53** | (-3.77, 0.49) | -11.48 | (-26.06, -3.15) | -5.77 | (-16.20, -2.27) |
| | | 160 | **-0.10** | (-3.68, 0.74) | -2.91 | (-5.48, -0.91) | -1.45 | (-3.72, 0.12) |
| | | 240 | **-0.59** | (-5.44, 0.68) | -1.24 | (-2.21, -0.22) | -0.71 | (-1.43, 0.21) |
| | | 320 | -0.41 | (-3.57, 0.68) | -0.43 | (-1.51, 0.21) | **-0.23** | (-1.15, 0.35) |

**Baseline Methods.** We evaluate the following two baseline methods for adapting the MDN. 1) A new MDN is initialized using the weights of the MDN trained on the source dataset, and it is adapted using the target dataset. 2) Same as baseline 1, but only the weights of the final layer are adapted (fine-tuned) using the target dataset. The above methods are referred to as **transfer** and **transfer-last-layer** respectively. We used the Adam optimization method (Kingma & Ba, 2015) for 200 epochs, with a batch size of 10 or 0.01 times the target dataset size, whichever is larger. Table 2 compares the number of parameters being optimized by the proposed and baseline MDN adaptation

methods for the architecture in Table 4. We observe that the number of parameters optimized by the proposed method is orders of magnitude smaller.

**Results and Inference.** Since the MDN is a generative model, we evaluate the conditional log-likelihood (CLL) of the adapted Gaussian mixture on an unseen test set of 25000 samples from the target channel. Table 1 compares the median and $95\%$ confidence interval (CI) of the CLL for three (source, target) channel pairs. For each pair, the methods are run on 50 randomly generated training, adaptation, and test datasets. The training dataset is sampled from the source channel, while the adaptation and test datasets are sampled from the target channel. The SNR of the source and target channels are independently and randomly selected from the range $10\,\mathrm{dB}$ to $20\,\mathrm{dB}$ for each trial. We observe that the proposed method has a higher median CLL in a majority of the cases (particularly for low sample sizes), and has comparable median CLL for higher sample sizes. We also observe that the baseline methods have a more negatively-skewed CLL at the smaller sample sizes, suggesting that they often found poor adaptation solutions.

We performed additional experiments where the source and target channel distributions are Gaussian mixtures with different parameters that are randomly sampled (in a controlled way). We also evaluated the performance of the methods in the special cases where i) the source and target distributions are the same, and ii) where the number of components in the two mixtures is mis-matched. These results in Table 5 in Appendix D support the strong adaptation performance of the proposed method.

Table 2: Number of parameters being optimized by the MDN adaptation methods.

| Adaptation method | # parameters | # parameters (specific) |
|---|---|---|
| Transfer | $n_h\,(n_h + d + 2)$ $+\,k\,(2\,d + 1)\,(n_h + 1)$ | 12925 |
| Transfer-last-layer | $k\,(2\,d + 1)\,(n_h + 1)$ | 2525 |
| Proposed | $k\,(d^2 + 2\,d + 2)$ | 50 |

## 5.2 AUTOENCODER ADAPTATION ON SIMULATED CHANNELS

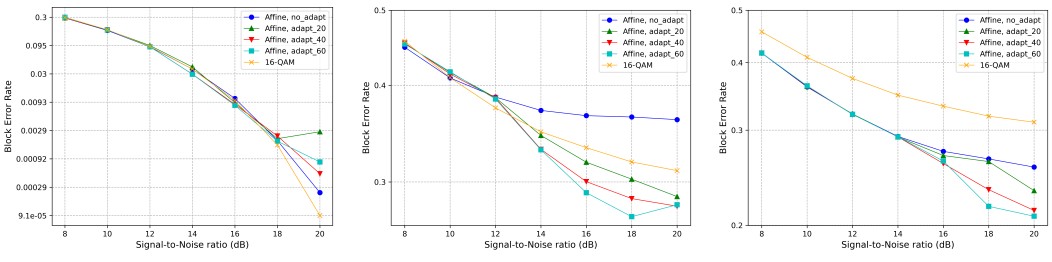

(a) AWGN to Ricean fading.   (b) AWGN to Uniform fading.   (c) Ricean fading to Uniform fading.

Figure 4: Results of affine transformation based adaptation on simulated channels.

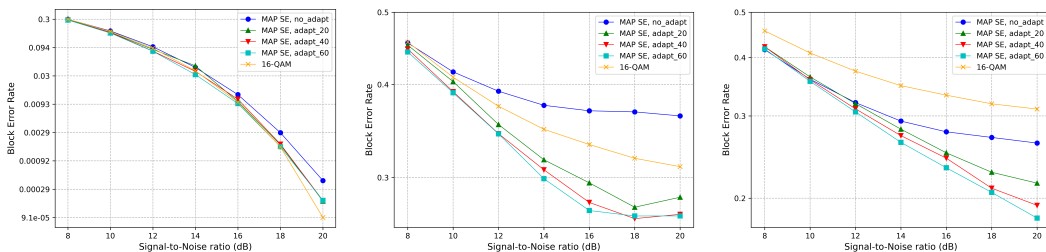

(a) AWGN to Ricean fading.   (b) AWGN to Uniform fading.   (c) Ricean fading to Uniform fading.

Figure 5: Results of MAP SE based adaptation on simulated channels.

We evaluate the proposed decoder adaptation methods on different pairs of simulated source and target channel distributions. The setup for this experiment for adapting from a source channel A to a target channel B is as follows. The autoencoder is initially trained using data from the source channel A at an SNR of $14\,\mathrm{dB}$. Details of how the SNR is related to the distribution parameters of the simulated channels is discussed in Appendix E. The MDN and the decoder are adapted using a small dataset from the target channel B for different fixed SNRs varied over $8\,\mathrm{dB}$ to $20\,\mathrm{dB}$ in steps of $2\,\mathrm{dB}$. For each SNR, the adaptation is repeated over 10 randomly-sampled datasets from the target channel, and the average block error rate (BLER) values are calculated on a large held-out test dataset (specific to each SNR). The size of training dataset (from channel A) and test dataset (from channel

B) are both set to $20,000$ samples per symbol, with 16 symbols. The size of the adaptation dataset from the target channel B is varied over $20, 40,$ and $60$ samples per symbol.

The results of this experiment are given in Figs. 4 and 5 for three pairs of source and target channels. Figure 4 corresponds to the adaptation method of § 4.1 referred to as **Affine**, and Figure 5 corresponds to the adaptation method of § 4.2 referred to as **MAP SE**. The plots show the BLER vs. SNR curve, with average BLER on the y-axis (log-scaled) and SNR on the x-axis. The performance of a standard 16-QAM decoder (Haykin, 1988), and an autoencoder trained on the source channel without any adaptation (referred to as **no_adapt**) are included as baselines. The number of samples per symbol from the target channel used by the proposed methods is shown as a suffix to the method name. In Appendix D.3, we compare the proposed adaptation methods to the best-case performance of a fully-retrained autoencoder.

**Observations and Takeaways. 1)** Both the adaptation methods significantly decrease the BLER for the cases AWGN to Uniform fading and Ricean fading to Uniform fading. **2)** For the case of AWGN to Ricean fading, the adaptation methods perform at the same level or slightly worse compared to the baselines. We think this is because the distribution of the two domains are not very different. **3)** In general, the BLER decreases with increasing number of the target domain samples. **4)** Between the two adaptation methods, MAP SE generally outperforms than the Affine method (see Appendix D.4).

### 5.3 AUTOENCODER ADAPTATION ON REAL FPGA TRACES

We evaluate the performance of the adaptation methods on real over-the-air wireless experiments. We use a recent high-performance mmWave testbed (Lacruz et al., 2021), featuring a high-end FPGA board with $2\,\text{GHz}$ bandwidth per-channel and $60\,\text{GHz}$ SIVERS antennas (SIVER-SIMA, 2020). We train the MDN with a standard 16-QAM constellation using $96,000$

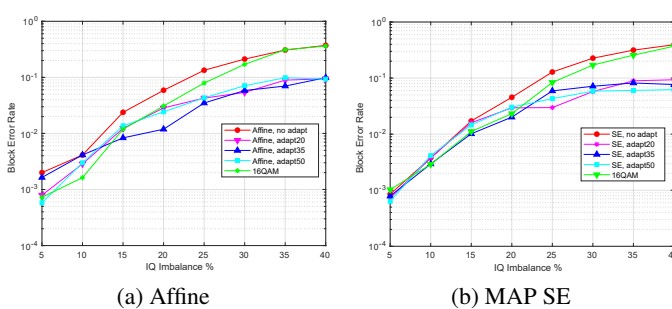

(a) Affine        (b) MAP SE

Figure 6: Results of adaptation on the real FPGA traces.

samples. We introduce distribution changes via IQ imbalance based distortions to the symbol constellation, and gradually increase the level of imbalance to the system [4]. More details on the experimental setup and the source and target domains are given in Appendix D.2. We evaluate the performance of the proposed adaptation for 20, 35 and 50 samples per symbol. The BLER of the proposed adaptation methods and the baseline methods (16-QAM and no adaptation) is shown as a function of the IQ imbalance in Fig. 6. The proposed methods (both Affine and MAP SE) show an order of magnitude decrease in BLER compared to the baselines when the IQ imbalance is over 25%.

## 6 CONCLUSIONS

In this paper we proposed a fast and light-weight method for adapting a Gaussian MDN with very limited number of samples from the target distribution. We applied the MDN adaptation to an autoencoder-based e2e communication system, specifically by transforming the inputs to the decoder such that their class-conditional distributions are close to that of the source domain. This allows for fast adaptation of both the MDN channel and the autoencoder without the need for expensive data collection and retraining. We demonstrated the effectiveness of the proposed methods through extensive experiments on both simulated channels and a real mmWave FPGA testbed.

**Limitations & Future Work.** The proposed adaptation for a Gaussian MDN is primarily targeted for low-dimensional problems such as the wireless channel. It can be challenging to apply on high-dimensional input domains with structure. Extensions of the proposed work to deep generative models based on normalizing flows (Dinh et al., 2017; Kingma & Dhariwal, 2018; Weng, 2018) is an interesting direction, which would be more suitable for high-dimensional inputs. In this work, we do not adapt the encoder network, *i.e.*, the autoencoder constellation is not adapted to changes in the channel distribution. Adapting the encoder, decoder, and channel networks jointly would allow for more flexibility, but would likely be slower and require more data from the target distribution.

---

[4]IQ imbalance is a common issue in RF communication that introduces distortions to the final constellation.

## ETHICS STATEMENT

This work does not raise any ethical issues to the best of our knowledge and judgement.

## REPRODUCIBILITY STATEMENT

The experimental results can be reproduced through our code base at `https://anonymous.4open.science/r/domain_adaptation-7C0D/`.

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

## APPENDIX

Table 3: Commonly used notations and definitions

| Notation | Description |
|---|---|
| $s \in \mathcal{S} := \{1, \cdots, m\}$ | Input message or class label. Usually $m = 2^b$, where $b$ is the number of bits. |
| $\mathbf{1}_{s,m}$ or simply $\mathbf{1}_s$, $s \in \mathcal{S}$ | One-hot-coded vector of a message $s$, with 1 at position $s$ and zeros elsewhere. |
| $\mathbf{x} \in \mathcal{X} \subset \mathbb{R}^d$ with $|\mathcal{X}| = m$ | Encoded representation or symbol vector corresponding to an input message. |
| $\mathbf{y} \in \mathbb{R}^d$ | Channel output that is the feature vector to be classified by the decoder. |
| $K \in \{1, \cdots, k\}$ | Categorical random variable denoting the mixture component of origin. |
| $\mathbf{E}_{\boldsymbol{\theta}_e}(\mathbf{1}_s)$ | Encoder NN with parameters $\boldsymbol{\theta}_e$ mapping a one-hot-coded message to a symbol vector in $\mathbb{R}^d$. |
| $\mathbf{D}_{\boldsymbol{\theta}_d}(\mathbf{y}) = [P_{\boldsymbol{\theta}_d}(1 \,|\, \mathbf{y}), \cdots, P_{\boldsymbol{\theta}_d}(m \,|\, \mathbf{y})]^T$ | Decoder NN with parameters $\boldsymbol{\theta}_d$ mapping the channel output into probabilities over the message set. |
| $\widehat{S}(\mathbf{y}) = \operatorname{argmax}_{s \in \mathcal{S}} P_{\boldsymbol{\theta}_d}(s \,|\, \mathbf{y})$ | MAP prediction of the input message by the decoder. |
| $P_{\boldsymbol{\theta}_c}(\mathbf{y} \,|\, \mathbf{x})$ | Conditional density (generative) model of the channel with parameters $\boldsymbol{\theta}_c$. |
| $\phi(\mathbf{x}) = \mathbf{M}_{\boldsymbol{\theta}_c}(\mathbf{x})$ | Mixture density network that predicts the parameters of a Gaussian mixture. |
| $\mathbf{y} = \mathbf{h}_{\boldsymbol{\theta}_c}(\mathbf{x}, \mathbf{z})$ | Transfer or sampling function corresponding to the channel conditional density. |
| $\mathbf{z} \in \mathbb{R}^\ell$ | Random vector independent of $\mathbf{x}$ that captures the stochasticity of the channel. |
| $\mathbf{f}_{\boldsymbol{\theta}}(\mathbf{1}_s) = \mathbf{D}_{\boldsymbol{\theta}_d}(\mathbf{h}_{\boldsymbol{\theta}_c}(\mathbf{E}_{\boldsymbol{\theta}_e}(\mathbf{1}_s), \mathbf{z}))$ | Input-output mapping of the autoencoder with combined parameter vector $\boldsymbol{\theta}^T = [\boldsymbol{\theta}_e^T, \boldsymbol{\theta}_c^T, \boldsymbol{\theta}_d^T]$. |
| $\boldsymbol{\psi}^T = [\boldsymbol{\psi}_1^T, \cdots, \boldsymbol{\psi}_k^T]$ | Affine transformation parameters per component used to adapt the MDN. |
| $\mathbf{g}_{\mathbf{x}i}$ and $\mathbf{g}_{\mathbf{x}i}^{-1}$, $i \in [k]$, $\mathbf{x} \in \mathcal{X}$ | Affine and inverse-affine transformations between the component densities of the Gaussian mixtures. |
| $D_{\mathrm{KL}}(p, q)$ | Kullback-Leibler divergence between the distributions $p$ and $q$. |
| $N(\cdot \,|\, \boldsymbol{\mu}, \boldsymbol{\Sigma})$ | Multivariate Gaussian density with mean vector $\boldsymbol{\mu}$ and covariance matrix $\boldsymbol{\Sigma}$. |
| $\mathrm{Cat}(p_1, \cdots, p_k)$ | Categorical distribution with $p_i \geq 0$ and $\sum_i p_i = 1$. |
| $\mathbb{1}(c)$ | Indicator function mapping a predicate $c$ to 1 if true and 0 if false. |
| $\|\mathbf{x}\|_p$ | $\ell_p$ norm of a vector $\mathbf{x}$. |

The appendices are organized as follows:

- Appendix A provides background on end-to-end learning of a communication autoencoder, MDN-based generative modeling, and domain adaptation.
- Appendix B provides details on the training and sampling (transfer) function of MDNs.
- Appendix C provides additional details on the proposed method that were omitted from the main paper. This includes:
  - Appendix C.1 discusses the feature and parameter transformations between multivariate Gaussians.
  - Appendix C.2 derives the KL divergence between the source and target Gaussian mixtures.
  - Appendix C.3 discusses the validation metric used for setting the hyper-parameter $\lambda$.
  - Appendix C.4 discusses the computational complexity of the proposed methods.
  - Appendix C.5 provides details on training the MAP symbol estimation autoencoder.
- Appendix D provides additional experimental results, including ablation studies of the proposed method.
- Appendix E provides details on the simulated channel models that were used in our experiments.

## A   BACKGROUND

In this section, we provide a background on the following topics: 1) components of an end-to-end autoencoder-based communication system, 2) generative modeling using mixture density networks, 3) loss function and training algorithm of the autoencoder, and 4) a primer on domain adaptation.

### A.1   AUTOENCODER-BASED END-TO-END LEARNING

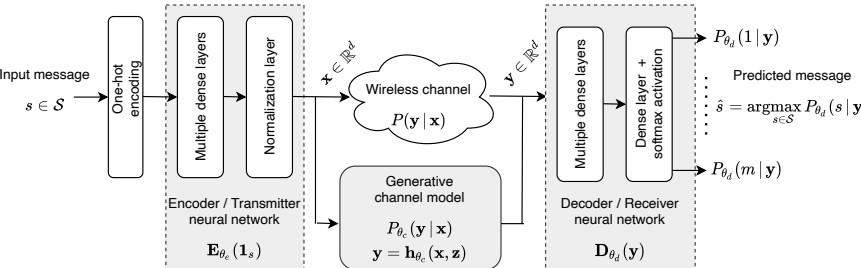

Figure 7: Representation of an end-to-end autoencoder-based communication system with a generative channel model.

Consider a single-input, single-output (SISO) wireless communication system as shown in Fig. 7, where the transmitter encodes and transmits messages from the set $\mathcal{S} = \{1, 2, \cdots, m\}$ to the receiver through $d \geq 2$ discrete uses of the wireless channel. The receiver attempts to accurately decode the transmitted message from the distorted and noisy channel output $\mathbf{y}$. We discuss the end-to-end learning of such a system using the concept of autoencoders (O'Shea & Hoydis, 2017; Dörner et al., 2018).

**Transmitter / Encoder Neural Network.** The transmitter or encoder part of the autoencoder is modeled as a multi-layer, feed-forward neural network (NN) that takes as input the one-hot-coded representation $\mathbf{1}_s$ of a message $s \in \mathcal{S}$, and produces an encoded symbol vector $\mathbf{x} = \mathbf{E}_{\boldsymbol{\theta}_e}(\mathbf{1}_s) \in \mathbb{R}^d$. Here, $\boldsymbol{\theta}_e$ is the parameter vector (weights and biases) of the encoder NN and $d$ is the encoding dimension. Due to hardware constraints present at the transmitter, a normalization layer is used as the final layer of the encoder network in order to constrain the average power and/or the amplitude of the symbol vectors. The average power constraint is defined as $\mathbb{E}[\|\mathbf{x}\|_2^2] = \mathbb{E}_S[\|\mathbf{E}_{\boldsymbol{\theta}_e}(\mathbf{1}_S)\|_2^2] \leq c$, where the expectation is over the prior distribution of the input messages, and $c$ is typically set to $1$. The amplitude constraint is defined as $|x_i| \leq 1, \ \forall i \in [d]$. The size of the message set is usually chosen to be a power of 2, *i.e.*, $m = 2^b$ representing $b$ bits of information. Following (O'Shea & Hoydis, 2017), the communication rate of this system is the number of bits transmitted per channel use, which in this case is $R = b \,/\, d$. An autoencoder transmitting $b$ bits over $d$ uses of the channel is referred to as a $(d, b)$ autoencoder. For example, a $(2, 4)$ autoencoder uses a message set of size 16 and an encoding dimension of 2, with a communication rate $R = 2$ bits/channel use.

**Receiver / Decoder Neural Network.** The receiver or decoder component is also a multilayer, feedforward NN that takes the channel output $\mathbf{y} \in \mathbb{R}^d$ as its input and outputs a probability distribution over the $m$ messages. The input-output mapping of the decoder NN can be expressed as $\mathbf{D}_{\boldsymbol{\theta}_d}(\mathbf{y}) := [P_{\boldsymbol{\theta}_d}(1 \,|\, \mathbf{y}), \cdots, P_{\boldsymbol{\theta}_d}(m \,|\, \mathbf{y})]^T$, where $\boldsymbol{\theta}_d$ is the parameter vector of the decoder NN. The softmax activation function is used at the final layer to ensure that the outputs are valid probabilities. The message corresponding to the highest output probability is predicted as the decoded message, *i.e.*, $\widehat{S}(\mathbf{y}) = \mathrm{argmax}_{s \in \mathcal{S}} \, P_{\boldsymbol{\theta}_d}(s \,|\, \mathbf{y})$. The decoder NN is essentially a *discriminative classifier* that learns to accurately categorize the received (distorted) symbol vector into one of the $m$ message classes. This is in contrast to conventional autoencoders, where the decoder learns to accurately reconstruct a high-dimensional tensor input from its low-dimensional representation learned by the encoder. The mean-squared and median-absolute error are commonly used end-to-end performance metrics for conventional autoencoders. In the case of communication autoencoders, the *symbol or block error rate (BLER)*, defined as $\mathbb{E}_{S,\mathbf{Y}}[\mathbb{1}(\widehat{S}(\mathbf{Y}) \neq S)]$, is used as the end-to-end performance metric.

**Channel Model.** As discussed in § 1, the wireless channel can be represented by a *conditional probability density* of the channel output given its input $P(\mathbf{y} \,|\, \mathbf{x})$. The channel can be equivalently characterized by a *stochastic transfer function* $\mathbf{y} = \mathbf{h}(\mathbf{x}, \mathbf{z})$ that transforms the encoded symbol vector into the channel output, where $\mathbf{z}$ captures the stochastic components of the channel (*e.g.*, random noise, phase offsets). For example, an additive white Gaussian noise (AWGN) channel is represented by $\mathbf{y} = \mathbf{h}(\mathbf{x}, \mathbf{z}) = \mathbf{x} + \mathbf{z}$, with $\mathbf{z} \sim \mathcal{N}(\cdot \,|\, \mathbf{0}, \sigma^2 \, \mathbf{I}_d)$ and $P(\mathbf{y} \,|\, \mathbf{x}) = \mathcal{N}(\mathbf{y} \,|\, \mathbf{x}, \sigma^2 \, \mathbf{I}_d)$. For realistic wireless channels, the transfer function and conditional probability density are usually unknown and hard to approximate well with standard mathematical models. Recently, a number of works have applied generative models such as conditional generative adversarial networks (GANs) (O'Shea et al., 2019; Ye et al., 2018), MDNs (García Martí et al., 2020), and conditional variational autoencoders (VAEs) (Xia et al., 2020) for modeling the wireless channel. To model a wireless channel, generative methods learn a parametric model $P_{\boldsymbol{\theta}_c}(\mathbf{y} \,|\, \mathbf{x})$ (possibly a neural network) that closely approximates the true conditional density of the channel from a dataset of channel input, output observations. Learning a generative model of the channel comes with *important advantages*. 1) Once the parameters of the channel model are learned from data, the model can be used to generate any number of representative samples from the channel distribution. 2) A channel model with a differentiable transfer function makes it possible to backpropagate gradients of the autoencoder loss function through the channel and train the autoencoder using stochastic gradient descent (SGD)-based optimization. 3) It allows for continuous adaptation of the generative channel model to variations in the channel conditions, and thereby maintain a low BLER of the autoencoder.

## A.2 Generative Channel Model using a Mixture Density Network

In this work, we use an MDN (Bishop, 1994; 2007) with Gaussian components to model the conditional density of the channel output given its input. MDNs can model complex conditional densities by combining a (feed-forward) neural network with a standard parametric mixture model (*e.g.*, mixture of Gaussians). The MDN learns to predict the parameters of the mixture model $\phi(\mathbf{x})$ as a function of the channel input $\mathbf{x}$. This can be expressed as $\phi(\mathbf{x}) = \mathbf{M}_{\boldsymbol{\theta}_c}(\mathbf{x})$, where $\boldsymbol{\theta}_c$ is the parameter vector (weights and biases) of the neural network. The parameters of the mixture model defined by the MDN are a concatenation of the parameters from the $k$ density components, *i.e.*, $\phi(\mathbf{x})^T = [\phi_1(\mathbf{x})^T, \cdots, \phi_k(\mathbf{x})^T]$, where $\phi_i(\mathbf{x})$ is the parameter vector of component $i$. Focusing on a Gaussian mixture, the *channel conditional density* modeled by the MDN is given by

$$P_{\boldsymbol{\theta}_c}(\mathbf{y} \,|\, \mathbf{x}) \;=\; \sum_{i=1}^{k} P_{\boldsymbol{\theta}_c}(K = i \,|\, \mathbf{x}) \, P_{\boldsymbol{\theta}_c}(\mathbf{y} \,|\, \mathbf{x}, K = i) \;=\; \sum_{i=1}^{k} \pi_i(\mathbf{x}) \, N(\mathbf{y} \,|\, \boldsymbol{\mu}_i(\mathbf{x}), \boldsymbol{\sigma}_i^2(\mathbf{x})), \quad (13)$$

where $\boldsymbol{\mu}_i(\mathbf{x}) \in \mathbb{R}^d$ is the mean vector, $\boldsymbol{\sigma}_i^2(\mathbf{x}) \in \mathbb{R}_+^d$ is the variance vector, and $\pi_i(\mathbf{x}) \in [0, 1]$ is the weight (prior probability) of component $i$. Also, $K$ is the latent random variable denoting the mixture component of origin. We have assumed that the Gaussian components have a diagonal covariance matrix, with $\boldsymbol{\sigma}_i^2(\mathbf{x})$ being the diagonal elements [5]. The weights of the mixture are parameterized using the softmax function as $\pi_i(\mathbf{x}) = e^{\alpha_i(\mathbf{x})} / \sum_{j=1}^{k} e^{\alpha_j(\mathbf{x})}$, $\forall i$ in order to satisfy the probability constraint. The MDN simply predicts the un-normalized weights $\alpha_i(\mathbf{x}) \in \mathbb{R}$ (also known as the *prior logits*). For a Gaussian MDN, the parameter vector of component $i$ is defined as $\phi_i(\mathbf{x})^T = [\alpha_i(\mathbf{x}), \boldsymbol{\mu}_i(\mathbf{x})^T, \boldsymbol{\sigma}_i^2(\mathbf{x})^T]$, and its output parameter vector $\phi(\mathbf{x})$ has dimension $k\,(2d + 1)$. Details on the *conditional log-likelihood (CLL)* training objective and the *transfer function* of the MDN, including a differentiable approximation of the transfer function, are discussed in Appendix B.

## A.3 Loss Function and Training of the Autoencoder

In this section, we provide a formal discussion of the end-to-end training of the autoencoder. First, let us define the input-output mapping of the autoencoder as $\mathbf{f}_{\boldsymbol{\theta}}(\mathbf{1}_s) = \mathbf{D}_{\boldsymbol{\theta}_d}(\mathbf{h}_{\boldsymbol{\theta}_c}(\mathbf{E}_{\boldsymbol{\theta}_e}(\mathbf{1}_s), \mathbf{z})) = (\mathbf{D}_{\boldsymbol{\theta}_d} \circ \mathbf{h}_{\boldsymbol{\theta}_c}(\cdot, \mathbf{z}) \circ \mathbf{E}_{\boldsymbol{\theta}_e})(\mathbf{1}_s)$, where $\boldsymbol{\theta}^T = [\boldsymbol{\theta}_e^T, \boldsymbol{\theta}_c^T, \boldsymbol{\theta}_d^T]$ is the combined vector of parameters from the encoder, channel, and decoder. Given an input message $s \in \mathcal{S}$, the autoencoder maps the one-hot-coded representation of $s$ into an output probability vector over the message set. Note that, while the encoder and decoder neural networks are deterministic, a forward pass through the autoencoder is stochastic due to the channel transfer function $\mathbf{h}_{\boldsymbol{\theta}_c}$. The learning objective of the autoencoder is to accurately recover the input message at the decoder with a high probability. The cross-entropy (CE) loss, which is commonly used for training classifiers, is also suitable for end-to-end training of the autoencoder. For an input $s$ with encoded representation $\mathbf{x} = \mathbf{E}_{\boldsymbol{\theta}_e}(\mathbf{1}_s)$, channel output $\mathbf{y} = \mathbf{h}_{\boldsymbol{\theta}_c}(\mathbf{x}, \mathbf{z})$, and decoded output $\mathbf{D}_{\boldsymbol{\theta}_d}(\mathbf{y}) = [P_{\boldsymbol{\theta}_d}(1 \,|\, \mathbf{y}), \cdots, P_{\boldsymbol{\theta}_d}(m \,|\, \mathbf{y})]^T$, the CE loss is given by

$$\begin{aligned} \ell_{\mathrm{CE}}(\mathbf{1}_s, \mathbf{f}_{\boldsymbol{\theta}}(\mathbf{1}_s)) &= -\mathbf{1}_s^T \log \mathbf{f}_{\boldsymbol{\theta}}(\mathbf{1}_s) = -\mathbf{1}_s^T \log \mathbf{D}_{\boldsymbol{\theta}_d}(\mathbf{h}_{\boldsymbol{\theta}_c}(\mathbf{E}_{\boldsymbol{\theta}_e}(\mathbf{1}_s), \mathbf{z})) \\ &= -\log P_{\boldsymbol{\theta}_d}(s \,|\, \mathbf{h}_{\boldsymbol{\theta}_c}(\mathbf{E}_{\boldsymbol{\theta}_e}(\mathbf{1}_s), \mathbf{z})), \end{aligned} \quad (14)$$

which is always non-negative and takes the minimum value 0 when the correct message is decoded with probability 1. The autoencoder aims to minimize the following expected CE loss over the input message set and the channel output:

$$\mathbb{E}[\ell_{\mathrm{CE}}(\mathbf{1}_S, \mathbf{f}_{\boldsymbol{\theta}}(\mathbf{1}_S))] \;=\; -\sum_{s=1}^{m} p(s) \int_{\mathbb{R}^d} P_{\boldsymbol{\theta}_c}(\mathbf{y} \,|\, \mathbf{E}_{\boldsymbol{\theta}_e}(\mathbf{1}_s)) \log P_{\boldsymbol{\theta}_d}(s \,|\, \mathbf{y}) \, d\mathbf{y}. \quad (15)$$

Here $p(s)$, $\forall s \in \mathcal{S}$ is the prior probability of the input messages, which is usually assumed to be uniform in the absence of prior knowledge. In practice, the autoencoder minimizes an empirical estimate of the expected CE loss function by generating a large set of samples from the channel conditional density given each message. Let $\mathcal{Y}^{(s)} = \{\mathbf{y}_n^{(s)} = \mathbf{h}_{\boldsymbol{\theta}_c}(\mathbf{E}_{\boldsymbol{\theta}_e}(\mathbf{1}_s), \mathbf{z}_n), \; n = 1, \cdots, N\}$ denote a set of independent and identically distributed (iid) samples from $P_{\boldsymbol{\theta}_c}(\mathbf{y} \,|\, \mathbf{E}_{\boldsymbol{\theta}_e}(\mathbf{1}_s))$, the

---

[5]The diagonal covariance assumption does not imply conditional independence of $\mathbf{y}$ as long as $k > 1$.

channel conditional density given message $s$. Also, let $\mathcal{Y} = \cup_s \mathcal{Y}^{(s)}$ denote the combined set of samples. The empirical expectation of the autoencoder CE loss (15) is then given by

$$\mathcal{L}_{auto}(\boldsymbol{\theta}\,;\mathcal{Y}) \;=\; -\sum_{s=1}^{m} p(s)\,\frac{1}{N}\sum_{n=1}^{N} \log P_{\boldsymbol{\theta}_d}(s\,|\,\mathbf{h}_{\boldsymbol{\theta}_c}(\mathbf{E}_{\boldsymbol{\theta}_e}(\mathbf{1}_s), \mathbf{z}_n)). \tag{16}$$

It is clear from the above equation that the channel transfer function $\mathbf{h}_{\boldsymbol{\theta}_c}$ should be differentiable in order to be able to backpropagate gradients through the channel to the encoder network. The transfer function defining sample generation for a Gaussian MDN channel is discussed in Appendix B.

The training algorithm for jointly learning the autoencoder and channel model (based on (García Martí et al., 2020)) is given in Algorithm 1. It is an alternating (cyclic) optimization of the channel parameters and the autoencoder (encoder and decoder) parameters. The reason this type of alternating optimization is required is because the empirical expectation of the CE loss Eq. (16) is valid *only when* the channel conditional density (*i.e.*, $\boldsymbol{\theta}_c$) is fixed. The training algorithm can be summarized as follows. First, the channel model is trained for $N_{ce}$ epochs using data sampled from the channel with an initial encoder constellation (*e.g.*, M-QAM). With the channel model parameters fixed, the parameters of the encoder and decoder networks are optimized for one epoch of mini-batch SGD updates (using any adaptive learning rate algorithm *e.g.*, Adam (Kingma & Ba, 2015)). Since the channel model is no longer optimal for the updated encoder constellation, it is retrained for $N_{ce}$ epochs using data sampled from the channel with the updated constellation. This alternate training of the encoder/decoder and the channel networks is repeated for $N_{ae}$ epochs or until convergence.

---

**Algorithm 1** End-to-end training of the autoencoder with a generative channel model

---

1: **Inputs:** Message size $m$; Encoding dimension $d$; Initial constellation $\{\mathbf{E}_0(\mathbf{1}_s),\ \forall s \in \mathcal{S}\}$; Number of optimization epochs for the autoencoder $N_{ae}$ and channel $N_{ce}$.
2: **Output:** Trained network parameters $\boldsymbol{\theta}_e, \boldsymbol{\theta}_c, \boldsymbol{\theta}_d$.
3: Initialize the encoder, channel, and decoder network parameters.
4: Sample training data $\mathcal{D}_c^{(0)}$ from the channel using the initial constellation.
5: Train the channel model for $N_{ce}$ epochs to minimize $\mathcal{L}_{ch}(\boldsymbol{\theta}_c\,;\mathcal{D}_c^{(0)})$.
6: **for** epoch $t = 1, \cdots, N_{ae}$:
7:     Freeze the channel model parameters $\boldsymbol{\theta}_c$.
8:     Perform a round of mini-batch SGD updates of $\boldsymbol{\theta}_e$ and $\boldsymbol{\theta}_d$ with respect to $\mathcal{L}_{auto}(\boldsymbol{\theta}\,;\mathcal{Y})$.
9:     Sample training data $\mathcal{D}_c^{(t)}$ from the channel with the updated constellation $\{\mathbf{E}_{\boldsymbol{\theta}_e}(\mathbf{1}_s),\ \forall s \in \mathcal{S}\}$.
10:    Train the channel model for $N_{ce}$ epochs to minimize $\mathcal{L}_{ch}(\boldsymbol{\theta}_c\,;\mathcal{D}_c^{(t)})$.
11: **Return** $\boldsymbol{\theta}_e, \boldsymbol{\theta}_c, \boldsymbol{\theta}_d$.

---

Finally, we observe some interesting nuances of the communication autoencoder learning task that is not common to other domains such as images. 1) The size of the input space is finite, equal to the number of distinct messages $m$. Because of the stochastic nature of the channel transfer function, the same input message results in a different autoencoder output each time. 2) There is theoretically no limit on the number of samples that can be generated for training and validating the autoencoder. These two factors make the autoencoder learning less susceptible to overfitting, that is a common pitfall with neural network training.

## A.4 A Primer on Domain Adaptation

We provide a brief review of domain adaptation (DA) and discuss the key differences of our problem setting from that of standard DA. In the traditional learning setting, training and test data are assumed to be sampled independently from the same distribution $P(\mathbf{x}, y)$, where $\mathbf{x}$ and $y$ are the input vector and target respectively [6]. In many real world settings, it can be hard or impractical to collect a large labeled dataset $\mathcal{D}_t^\ell$ for a *target domain* where the machine learning model (*e.g.*, a DNN classifier) is to be deployed. On the other hand, it is common to have access to a large unlabeled dataset $\mathcal{D}_t^u$

---

[6]The notation used in this section is different from the rest of the paper, but consistent with the statistical learning literature.

from the target domain, and a large labeled dataset $\mathcal{D}_s^\ell$ from a different but related *source domain* [7]. Both $\mathcal{D}_s^\ell$ and $\mathcal{D}_t^u$ are much larger than $\mathcal{D}_t^\ell$, and in most cases there is no labeled data from the target domain (referred to as unsupervised DA). For the target domain, the unlabeled dataset (and labeled dataset if any) are sampled from an unknown target distribution, *i.e.*, $\mathbf{x} \in \mathcal{D}_t^u \sim P_t(\mathbf{x})$ and $(\mathbf{x}, y) \in \mathcal{D}_t^\ell \sim P_t(\mathbf{x}, y)$. For the source domain, the labeled dataset is sampled from an unknown source distribution, *i.e.*, $(\mathbf{x}, y) \in \mathcal{D}_s^\ell \sim P_s(\mathbf{x}, y)$. The goal of DA is to leverage the available labeled and unlabeled datasets from the two domains to learn a predictor, denoted by the parametric function $\hat{y} = f_{\boldsymbol{\theta}}(\mathbf{x})$, such that the following risk function w.r.t the target distribution is minimized:

$$R_t[f_{\boldsymbol{\theta}}] \,=\, \mathbb{E}_{(\mathbf{x}, y) \sim P_t}[\ell(f_{\boldsymbol{\theta}}(\mathbf{x}), y)] \,=\, \sum_y \int_{\mathbf{x}} P_t(\mathbf{x}, y)\, \ell(f_{\boldsymbol{\theta}}(\mathbf{x}), y)\, d\mathbf{x},$$

where $\ell(\hat{y}, y)$ is a loss function that penalizes the prediction $\hat{y}$ for deviating from the true value $y$ (*e.g.*, cross-entropy or hinge loss). In a similar way, we can define the risk function w.r.t the source distribution $R_s[f_{\boldsymbol{\theta}}]$. A number of seminal works in DA theory (Ben-David et al., 2006; Blitzer et al., 2007; Ben-David et al., 2010) have studied this learning setting and provide bounds on $R_t[f_{\boldsymbol{\theta}}]$ in terms of $R_s[f_{\boldsymbol{\theta}}]$ and the divergence between source and target domain distributions. Motivated by this foundational theory, a number of recent works (Ganin & Lempitsky, 2015; Ganin et al., 2016; Long et al., 2018; Saito et al., 2018; Zhao et al., 2019; Johansson et al., 2019) have proposed using DNNs for adversarially learning a shared representation across the source and target domains such that a predictor using this representation and trained using labeled data from only the source domain also generalizes well to the target domain. An influential work in this line of DA is the domain adversarial neural network (DANN) proposed by (Ganin & Lempitsky, 2015) and later by (Ganin et al., 2016). The key idea behind the DANN approach is to adversarially train a label predictor NN and a domain discriminator NN in order to learn a feature representation for which i) the source and target inputs are nearly indistinguishable to the domain discriminator, and ii) the label predictor has good generalization performance on the source domain inputs.

**Special Cases of DA.** While the general DA problem addresses the scenario where $P_s(\mathbf{x}, y)$ and $P_t(\mathbf{x}, y)$ are different, certain special cases of DA have also been explored. One such special case is *covariate shift* (Sugiyama et al., 2007; Sugiyama & Kawanabe, 2012), where only the marginal distribution of the inputs changes (*i.e.*, $P_t(\mathbf{x}) \neq P_s(\mathbf{x})$), but the conditional distribution of the target given the input does not change (*i.e.*, $P_t(y \,|\, \mathbf{x}) \approx P_s(y \,|\, \mathbf{x})$). Another special case is the so-called *label shift* or class-prior mismatch (Saerens et al., 2002; Du Plessis & Sugiyama, 2014), where only the marginal distribution of the label changes (*i.e.*, $P_t(y) \neq P_s(y)$), but the conditional distribution of the input given the target does not change (*i.e.*, $P_t(\mathbf{x} \,|\, y) \approx P_s(\mathbf{x} \,|\, y)$). Prior works have proposed targeted theory and methods for these special cases of DA.

## B  MDN TRAINING AND SAMPLE GENERATION

In this section, we provide details on the MDN training, followed by a discussion on the sampling function of an MDN, and how to make the sampling function differentiable to enable backpropagation-based training of the autoencoder. Given a dataset of input-output pairs sampled from the channel $\mathcal{D}_c = \{(\mathbf{x}_n, \mathbf{y}_n),\ n = 1, \cdots, N_c\}$, the MDN is trained to minimize the negative conditional log-likelihood (CLL) of the data given by

$$\mathcal{L}_{\text{ch}}(\boldsymbol{\theta}_c\,;\mathcal{D}_c) \,=\, -\frac{1}{N_c} \sum_{n=1}^{N_c} \log P_{\boldsymbol{\theta}_c}(\mathbf{y}_n \,|\, \mathbf{x}_n). \tag{17}$$

With a large $N_c$, the MDN can learn a sufficiently-complex parametric density model of the channel. The negative CLL objective can be interpreted as the sample estimate of the Kullback-Leibler divergence between the true (unknown) conditional density $P(\mathbf{y} \,|\, \mathbf{x})$ and the conditional density modeled by the MDN $P_{\boldsymbol{\theta}_c}(\mathbf{y} \,|\, \mathbf{x})$. Therefore, minimizing the negative CLL finds the MDN parameters $\boldsymbol{\theta}_c$ that lead to a close approximation of the true conditional density. Standard SGD-based optimization methods such as Adam (Kingma & Ba, 2015) can be applied to find the MDN parameters $\boldsymbol{\theta}_c$ that (locally) minimize the negative CLL.

After the MDN is trained, new simulated samples from the channel distribution can be generated from the Gaussian mixture using the following stochastic sampling method:

---

[7]One could have multiple source domains in practice; we consider the single source domain setting.

1. Randomly select a channel input $\mathbf{x}$ from the categorical prior distribution $\{p(\mathbf{x}), \mathbf{x} \in \mathcal{X}\}$.
2. Randomly select a component $K = i$ according to the mixture weights $\{\pi_1(\mathbf{x}), \cdots, \pi_k(\mathbf{x})\}$.
3. Randomly sample $\mathbf{z}$ from the standard $d$-dimensional Gaussian density $\mathbf{z} \sim N(\cdot \,|\, \mathbf{0}, \mathbf{I}_d)$.
4. Generate the channel output as $\mathbf{y} = \boldsymbol{\sigma}_i^2(\mathbf{x}) \odot \mathbf{z} + \boldsymbol{\mu}_i(\mathbf{x})$.

Recall that $\odot$ refers to the element-wise product of two vectors. The channel transfer or sampling function for a Gaussian MDN can thus be expressed as

$$\mathbf{y} = \mathbf{h}_{\boldsymbol{\theta}_c}(\mathbf{x}, \mathbf{z}) = \sum_{i=1}^{k} \mathbb{1}(K = i)\, (\boldsymbol{\sigma}_i^2(\mathbf{x}) \odot \mathbf{z} + \boldsymbol{\mu}_i(\mathbf{x})), \tag{18}$$

where $K \sim \text{Cat}(\pi_1(\mathbf{x}), \cdots, \pi_k(\mathbf{x}))$ and $\mathbf{z} \sim N(\cdot \,|\, \mathbf{0}, \mathbf{I}_d)$. Note that this transfer function is not differentiable w.r.t parameters $\pi_i(\mathbf{x})$ and the MDN weights predicting it, because of the indicator function. As such, it is not directly suitable for SGD (backpropagation) based end-to-end training of the autoencoder. We next propose a differentiable approximation of the MDN transfer function based on the Gumbel softmax reparametrization (Jang et al., 2017), which is used in our autoencoder implementation.

### B.1 DIFFERENTIABLE MDN TRANSFER FUNCTION

Consider the transfer function of the MDN in Eq. (18). We would like to replace sampling from the categorical mixture prior $\text{Cat}(\pi_1(\mathbf{x}), \cdots, \pi_k(\mathbf{x}))$ with a differentiable function that closely approximates it. We apply the Gumbel-Softmax reparametrization (Jang et al., 2017) which solves this exact problem. First, recall that the component prior probabilities can be expressed in terms of the prior logits as:

$$\pi_i(\mathbf{x}) = \frac{e^{\alpha_i(\mathbf{x})}}{\sum_{j=1}^{k} e^{\alpha_j(\mathbf{x})}}, \quad \forall i \in [k].$$

Consider $k$ iid standard Gumbel random variables $G_1, \cdots, G_k \overset{\text{iid}}{\sim} \text{Gumbel}(0, 1)$. It can be shown that, for any $\mathbf{x} \in \mathcal{X}$, the random variable

$$S(\mathbf{x}) = \underset{i \in [k]}{\operatorname{argmax}}\, G_i + \alpha_i(\mathbf{x}) \tag{19}$$

follows the categorical distribution $\text{Cat}(\pi_1(\mathbf{x}), \cdots, \pi_k(\mathbf{x}))$. This standard result is known as the Gumbel-max transformation. While Eq. (19) can be directly used inside the indicator function in Eq. (18), the $\operatorname{argmax}$ will still result in the transfer function being non-differentiable. Therefore, we use the following temperature-scaled softmax function as a smooth approximation of the $\operatorname{argmax}$

$$\widehat{S}_i(\mathbf{x}\,;\tau) = \frac{\exp[(G_i + \alpha_i(\mathbf{x}))\,/\,\tau]}{\sum_{j=1}^{k} \exp[(G_j + \alpha_j(\mathbf{x}))\,/\,\tau]}, \quad \forall i \in [k], \tag{20}$$

where $\tau > 0$ is a temperature constant. For small values of $\tau$, the temperature-scaled softmax will closely approximate the $\operatorname{argmax}$, and the vector $[\widehat{S}_1(\mathbf{x}\,;\tau), \cdots, \widehat{S}_k(\mathbf{x}\,;\tau)]$ will closely approximate the one-hot vector $[\mathbb{1}(S(\mathbf{x}) = 1), \cdots, \mathbb{1}(S(\mathbf{x}) = k)]$.

Applying this Gumbel softmax reparametrization in Eq. (18), we define a modified differentiable transfer function for the Gaussian MDN as

$$\mathbf{y} = \widehat{\mathbf{h}}_{\boldsymbol{\theta}_c}(\mathbf{x}, \mathbf{z}) = \sum_{i=1}^{k} \widehat{S}_i(\mathbf{x}\,;\tau)\, (\boldsymbol{\sigma}_i^2(\mathbf{x}) \odot \mathbf{z} + \boldsymbol{\mu}_i(\mathbf{x})). \tag{21}$$

With this transfer function, it is straightforward to compute gradients with respect to the prior logits $\alpha_i(\mathbf{x})$, $\forall i$. Another neat outcome of this approach is that the stochastic components (Gumbel random variables $G_i$) are fully decoupled from the deterministic parameters $\alpha_i(\mathbf{x})$ in the gradient calculations with respect to $\widehat{S}_i(\mathbf{x}\,;\tau)$. In our experiments, we used this Gumbel-softmax based smooth transfer function while training the autoencoder, but during prediction (inference), we use the exact $\operatorname{argmax}$ based transfer function. We found $\tau = 0.01$ to be a good choice for all the experiments.

## C ADDITIONAL DETAILS ON THE PROPOSED METHOD

In this section we provide additional details on the proposed method that were omitted from the main paper (sections 3 and 4).

## C.1 TRANSFORMATION BETWEEN MULTIVARIATE GAUSSIANS

We discuss the feature and parameter transformations between any two multivariate Gaussians. This result was applied in § 3 to formulate the MDN adaptation. Consider first the standard transformation from $\mathbf{y} \sim N(\cdot \mid \boldsymbol{\mu}, \boldsymbol{\Sigma})$ to $\widehat{\mathbf{y}} \sim N(\cdot \mid \widehat{\boldsymbol{\mu}}, \widehat{\boldsymbol{\Sigma}})$ given by:

- Apply a whitening transformation $\mathbf{z} = \mathbf{D}^{-1/2} \mathbf{U}^T (\mathbf{y} - \boldsymbol{\mu})$ such that $\mathbf{z} \sim N(\cdot \mid \mathbf{0}, \mathbf{I})$.
- Transform $\mathbf{z}$ into the new Gaussian density using $\widehat{\mathbf{y}} = \widehat{\mathbf{U}} \widehat{\mathbf{D}}^{1/2} \mathbf{z} + \widehat{\boldsymbol{\mu}}$.

We have denoted the eigen-decomposition of the covariance matrices by $\boldsymbol{\Sigma} = \mathbf{U}\mathbf{D}\mathbf{U}^T$ and $\widehat{\boldsymbol{\Sigma}} = \widehat{\mathbf{U}}\widehat{\mathbf{D}}\widehat{\mathbf{U}}^T$, where $\mathbf{U}$ and $\widehat{\mathbf{U}}$ are the orthogonal eigenvector matrices, and $\mathbf{D}$ and $\widehat{\mathbf{D}}$ are the diagonal eigenvalue matrices. Combining the two steps, the overall transformation from $\mathbf{y}$ to $\widehat{\mathbf{y}}$ is given by

$$\widehat{\mathbf{y}} = \widehat{\mathbf{U}} \widehat{\mathbf{D}}^{1/2} \mathbf{D}^{-1/2} \mathbf{U}^T (\mathbf{y} - \boldsymbol{\mu}) + \widehat{\boldsymbol{\mu}}.$$

Suppose we define the matrix $\mathbf{C} = \widehat{\mathbf{U}} \widehat{\mathbf{D}}^{1/2} \mathbf{D}^{-1/2} \mathbf{U}^T$, then it is easily verified that the covariance matrices are related by $\widehat{\boldsymbol{\Sigma}} = \mathbf{C} \boldsymbol{\Sigma} \mathbf{C}^T$. In general, the mean vector and covariance matrix of any two Gaussians can be related by the following parameter transformations:

$$\widehat{\boldsymbol{\mu}} = \mathbf{A} \boldsymbol{\mu} + \mathbf{b} \quad \text{and} \quad \widehat{\boldsymbol{\Sigma}} = \mathbf{C} \boldsymbol{\Sigma} \mathbf{C}^T,$$

with parameters $\mathbf{A} \in \mathbb{R}^{d \times d}$, $\mathbf{b} \in \mathbb{R}^d$, and $\mathbf{C} \in \mathbb{R}^{d \times d}$. Substituting the above parameter transformations into the feature transformation, we get

$$\widehat{\mathbf{y}} = \mathbf{C} (\mathbf{y} - \boldsymbol{\mu}) + \mathbf{A} \boldsymbol{\mu} + \mathbf{b}.$$

## C.2 DIVERGENCE BETWEEN THE SOURCE AND TARGET GAUSSIAN MIXTURES

Referring to § 3.2, we provide a detailed derivation of the KLD between the source and target Gaussian mixtures under the assumption of one-to-one association between the components.

$$
\begin{aligned}
\overline{D}(P_{\boldsymbol{\theta}_c}, P_{\widehat{\boldsymbol{\theta}}_c}) &= \mathbb{E}_{P_{\boldsymbol{\theta}_c}} \left[ \log \frac{P_{\boldsymbol{\theta}_c}(\mathbf{y}, K \mid \mathbf{x})}{P_{\widehat{\boldsymbol{\theta}}_c}(\mathbf{y}, K \mid \mathbf{x})} \right] \\
&= \sum_{\mathbf{x} \in \mathcal{X}} p(\mathbf{x}) \sum_{i=1}^{k} \int_{\mathbb{R}^d} P_{\boldsymbol{\theta}_c}(\mathbf{y}, K = i \mid \mathbf{x}) \log \frac{P_{\boldsymbol{\theta}_c}(\mathbf{y}, K = i \mid \mathbf{x})}{P_{\widehat{\boldsymbol{\theta}}_c}(\mathbf{y}, K = i \mid \mathbf{x})} \, d\mathbf{y} \\
&= \sum_{\mathbf{x} \in \mathcal{X}} p(\mathbf{x}) \sum_{i=1}^{k} P_{\boldsymbol{\theta}_c}(K = i \mid \mathbf{x}) \int_{\mathbb{R}^d} P_{\boldsymbol{\theta}_c}(\mathbf{y} \mid \mathbf{x}, K = i) \log \frac{P_{\boldsymbol{\theta}_c}(K = i \mid \mathbf{x}) \, P_{\boldsymbol{\theta}_c}(\mathbf{y} \mid \mathbf{x}, K = i)}{P_{\widehat{\boldsymbol{\theta}}_c}(K = i \mid \mathbf{x}) \, P_{\widehat{\boldsymbol{\theta}}_c}(\mathbf{y} \mid \mathbf{x}, K = i)} \, d\mathbf{y} \\
&= \sum_{\mathbf{x} \in \mathcal{X}} p(\mathbf{x}) \sum_{i=1}^{k} \pi_i(\mathbf{x}) \int_{\mathbb{R}^d} N\left(\mathbf{y} \mid \boldsymbol{\mu}_i(\mathbf{x}), \boldsymbol{\sigma}_i^2(\mathbf{x})\right) \left( \log \frac{\pi_i(\mathbf{x})}{\widehat{\pi}_i(\mathbf{x})} + \log \frac{N\left(\mathbf{y} \mid \boldsymbol{\mu}_i(\mathbf{x}), \boldsymbol{\sigma}_i^2(\mathbf{x})\right)}{N\left(\mathbf{y} \mid \widehat{\boldsymbol{\mu}}_i(\mathbf{x}), \widehat{\boldsymbol{\sigma}}_i^2(\mathbf{x})\right)} \right) d\mathbf{y} \\
&= \sum_{\mathbf{x} \in \mathcal{X}} p(\mathbf{x}) \sum_{i=1}^{k} \pi_i(\mathbf{x}) \log \frac{\pi_i(\mathbf{x})}{\widehat{\pi}_i(\mathbf{x})} \\
&\quad + \sum_{\mathbf{x} \in \mathcal{X}} p(\mathbf{x}) \sum_{i=1}^{k} \pi_i(\mathbf{x}) \, D_{\mathrm{KL}}\left( N\left(\cdot \mid \boldsymbol{\mu}_i(\mathbf{x}), \boldsymbol{\sigma}_i^2(\mathbf{x})\right), N\left(\cdot \mid \widehat{\boldsymbol{\mu}}_i(\mathbf{x}), \widehat{\boldsymbol{\sigma}}_i^2(\mathbf{x})\right) \right).
\end{aligned}
\tag{22}
$$

The second term in the final expression involves the KLD between two Gaussian densities, which (for general covariances) is given by

$$
\begin{aligned}
D_{\mathrm{KL}}\left( N(\cdot \mid \boldsymbol{\mu}, \boldsymbol{\Sigma}), N(\cdot \mid \widehat{\boldsymbol{\mu}}, \widehat{\boldsymbol{\Sigma}}) \right) &= \frac{1}{2} \log \frac{\det(\widehat{\boldsymbol{\Sigma}})}{\det(\boldsymbol{\Sigma})} + \frac{1}{2} \operatorname{tr}(\widehat{\boldsymbol{\Sigma}}^{-1} \boldsymbol{\Sigma}) \\
&\quad + \frac{1}{2} (\widehat{\boldsymbol{\mu}} - \boldsymbol{\mu})^T \widehat{\boldsymbol{\Sigma}}^{-1} (\widehat{\boldsymbol{\mu}} - \boldsymbol{\mu}) - \frac{d}{2}.
\end{aligned}
$$

Applying this result to the KLD term in Eq. (22), which has diagonal covariances, we get

$$D_{\mathrm{KL}}\left(N\left(\cdot \,|\, \boldsymbol{\mu}_i(\mathbf{x}), \boldsymbol{\sigma}_i^2(\mathbf{x})\right), N\left(\cdot \,|\, \widehat{\boldsymbol{\mu}}_i(\mathbf{x}), \widehat{\boldsymbol{\sigma}}_i^2(\mathbf{x})\right)\right)$$

$$= \frac{1}{2} \sum_{j=1}^{d} \left[\log c_{ij}^2 \;+\; \frac{1}{c_{ij}^2} \;+\; \frac{1}{c_{ij}^2 \, \sigma_{ij}^2(\mathbf{x})} \left(a_{ij}\,\mu_{ij}(\mathbf{x}) \;+\; b_{ij} \;-\; \mu_{ij}(\mathbf{x})\right)^2\right] - \frac{d}{2}. \qquad (23)$$

The other term in Eq. (22) involving the KLD between the component prior probabilties can be expressed as a function of the adaptation parameters $[\beta_1, \gamma_1, \cdots, \beta_k, \gamma_k]$ as follows:

$$\sum_{i=1}^{k} \pi_i(\mathbf{x}) \log \frac{\pi_i(\mathbf{x})}{\widehat{\pi}_i(\mathbf{x})} \;=\; \sum_{i=1}^{k} \frac{e^{\alpha_i(\mathbf{x})}}{z(\mathbf{x})} \left[\log \frac{e^{\alpha_i(\mathbf{x})}}{z(\mathbf{x})} \;-\; \log \frac{e^{\beta_i\,\alpha_i(\mathbf{x}) + \gamma_i}}{\widehat{z}(\mathbf{x})}\right]$$

$$= \; \log\Big(\sum_{i=1}^{k} e^{\beta_i\,\alpha_i(\mathbf{x}) + \gamma_i}\Big) \;-\; \log\Big(\sum_{i=1}^{k} e^{\alpha_i(\mathbf{x})}\Big) \;+\; \sum_{i=1}^{k} \frac{e^{\alpha_i(\mathbf{x})}}{z(\mathbf{x})} \left(\alpha_i(\mathbf{x}) - \beta_i\,\alpha_i(\mathbf{x}) - \gamma_i\right), \quad (24)$$

where $z(\mathbf{x}) = \sum_{j=1}^{k} e^{\alpha_j(\mathbf{x})}$ and $\widehat{z}(\mathbf{x}) = \sum_{j=1}^{k} e^{\beta_j\,\alpha_j(\mathbf{x}) + \gamma_j}$ are the normalization terms in the softmax function. Substituting Eqs. (23) and (24) into the last step of Eq. (22) gives the KLD between the source and target distributions as a function of the adaptation parameters $\boldsymbol{\psi}$.

## C.3 Validation Metric and Selection of $\lambda$

The choice of $\lambda$ in the adaptation objective (Eqs. (7) and (8)) is crucial as it sets the amount of regularization most suitable for the target domain distribution. We propose a validation metric for selecting $\lambda$ based on the CLL of the inverse-affine-transformed target dataset with respect to the source mixture density. The reasoning is that, if the adaptation finds a solution $\boldsymbol{\psi}$ that is a good fit for the target dataset, then the inverse feature transformations based on that solution should produce a transformed target dataset that has a high CLL with respect to the source mixture density. The validation metric is the negative CLL of the inverse-transformed target dataset, given by

$$\mathcal{L}_{\mathrm{val}}(\boldsymbol{\psi}\,;\mathcal{D}_c^{(t)}) \;=\; \frac{-1}{N_c^{(t)}} \sum_{n=1}^{N_c^{(t)}} \log P_{\boldsymbol{\theta}_c}\big(\,\mathbf{g}_{\mathbf{x}_n^{(t)}\,i_n^{(t)}}^{-1}(\mathbf{y}_n^{(t)}) \,|\, \mathbf{x}_n^{(t)}\big). \qquad (25)$$

Here $i_n^{(t)}$ is the best component assignment for the sample $(\mathbf{x}_n^{(t)}, \mathbf{y}_n^{(t)})$, given by

$$i_n^{(t)} \;=\; \underset{i \in [k]}{\operatorname{argmax}}\; P_{\widehat{\boldsymbol{\theta}}_c}(K = i \,|\, \mathbf{x}_n^{(t)}, \mathbf{y}_n^{(t)}). \qquad (26)$$

The above equation is simply the maximum-a-posteriori (MAP) rule applied to the component posterior of the target Gaussian mixture defined as

$$P_{\widehat{\boldsymbol{\theta}}_c}(K = i \,|\, \mathbf{x}, \mathbf{y}) \;=\; \frac{\widehat{\pi}_i(\mathbf{x})\,N(\mathbf{y}\,|\,\widehat{\boldsymbol{\mu}}_i(\mathbf{x}), \widehat{\boldsymbol{\sigma}}_i^2(\mathbf{x}))}{\sum_{j=1}^{k} \widehat{\pi}_j(\mathbf{x})\,N(\mathbf{y}\,|\,\widehat{\boldsymbol{\mu}}_j(\mathbf{x}), \widehat{\boldsymbol{\sigma}}_j^2(\mathbf{x}))}, \quad \forall i \in [k].$$

Note that the validation metric (25) is based on the source Gaussian mixture (with parameters $\boldsymbol{\theta}_c$), but the MAP component assignment for each target domain sample Eq. (26) is based on the target Gaussian mixture (with parameters $\widehat{\boldsymbol{\theta}}_c$). The adaptation objective is minimized with $\lambda$ varied over a range of values, and in each case the adapted solution $\boldsymbol{\psi}$ is evaluated using the validation metric. The pair of $\lambda$ and $\boldsymbol{\psi}$ resulting in the smallest validation metric is chosen as the final adapted solution.

## C.4 Complexity Analysis

We provide an analysis of the computational complexity of the proposed adaptation methods.

**MDN Adaptation.**

The number of free parameters being optimized in the MDN adaptation objective (Eqs. 7 or 8) is given by $|\boldsymbol{\psi}| = k\,(d^2 + 2\,d + 2)$. This is much smaller than the number of parameters in a typical MDN, even considering only the final fully-connected layer (see Table 2 for a comparison). Each step of the BFGS optimization involves computing the objective function, its gradient, and an estimate

of its inverse Hessian. The cost of one step of BFGS can thus be expressed as $O(N_c^{(t)}\, k\, d^2\, |\boldsymbol{\psi}|^2)$. Suppose BFGS runs for a maximum of $T$ iterations and the optimization is repeated for $L$ values of $\lambda$, then the overall cost of adaptation is given by $O(L\, T\, N_c^{(t)}\, k\, d^2\, |\boldsymbol{\psi}|^2)$. Note that the optimization for different $\lambda$ values can be easily solved in parallel.

**Test-time adaptation at the decoder.**

The computational cost of the proposed adaptation methods (for processing a single input) at the decoder are only slightly different from each other. The affine transformation method of § 4.1 computes the posterior distribution $P_{\widehat{\boldsymbol{\theta}}_c}(\mathbf{x}, i \,|\, \mathbf{y})$ over the set of symbols and components (of size $m\,k$). Computation of each exponent factor in the posterior distribution requires $O(d^3)$ operations for the full-covariance case, and $O(d)$ operations for the diagonal covariance case. This corresponds to calculation of the log of the Gaussian density. Therefore, computation of the maximum of the posterior distribution requires $O\big(k\,m\,(k\,m\,d^3)\big) = O\big(k^2\,m^2\,d^3\big)$ operations for the full-covariance case (and similarly for the diagonal case).

The MAP symbol estimation method of § 4.2 computes the posterior distribution $P_{\widehat{\boldsymbol{\theta}}_c}(\mathbf{x} \,|\, \mathbf{y})$ over the set of symbols (of size $m$). Computation of each exponent factor in the posterior distribution requires $O(k\,d^3)$ operations for the full covariance case, and $O(k\,d)$ operations for the diagonal covariance case. This corresponds to calculation of the log of the Gaussian mixture density with $k$ components. Therefore, computation of the maximum of this posterior distribution requires $O\big(m\,(m\,k\,d^3)\big) = O\big(k\,m^2\,d^3\big)$ operations for the full-covariance case.

Therefore, the computational complexity of the affine transformation method is larger than that of the MAP SE method by a factor of $k$. Usually $k$ is not large. However, the computational cost of both methods can be relatively high for large modulation orders (*e.g.*, $m = 256$ or $1024$ symbols).

A key advantage of the proposed adaptation is that it is very computationally efficient to implement at the receiver end of a communication system. Typically, $d$ is very small for communication problems, and $d = 2$ coincides with standard modulation techniques (*e.g.*, QPSK). The number of symbols or messages $m$ can vary from $4$ to $1024$ in powers of $2$. The number mixture components $k$ can be any positive integer, but is usually not more than a few tens to keep the size of the MDN practical.

### C.5 MAP Symbol Estimation Autoencoder

In this section, we discuss some details of the adapted decoder with MAP symbol estimation that were not addressed in § 4.2. The MAP SE method for adapting the decoder introduced the following transformation layer prior to the decoder in order to estimate the channel input $\mathbf{x}$ from the channel output $\mathbf{y}$:

$$\widehat{\mathbf{X}}_{\mathrm{map}}(\mathbf{y}) \;=\; \underset{\mathbf{x}\in\mathcal{X}}{\arg\max}\; P_{\boldsymbol{\theta}_c}(\mathbf{x}\,|\,\mathbf{y}) \;=\; \underset{\mathbf{x}\in\mathcal{X}}{\arg\max}\; \log P_{\boldsymbol{\theta}_c}(\mathbf{y}\,|\,\mathbf{x}) \;+\; \log p(\mathbf{x}).$$

As discussed in § 4.2, the presence of the non-differentiable $\arg\max$ poses a problem for backpropagation-based training of the autoencoder. We propose to address this using a temperature-scaled softmax approximation to the $\arg\max$, similar to the method discussed in Appendix B.1.

Consider the posterior distribution of the channel input given the channel output

$$P_{\boldsymbol{\theta}_c}(\mathbf{x}\,|\,\mathbf{y}) \;=\; \frac{p(\mathbf{x})\,P_{\boldsymbol{\theta}_c}(\mathbf{y}\,|\,\mathbf{x})}{\sum_{\mathbf{x}'\in\mathcal{X}} p(\mathbf{x}')\,P_{\boldsymbol{\theta}_c}(\mathbf{y}\,|\,\mathbf{x}')} \;=\; \frac{\exp(q_{\boldsymbol{\theta}_c}(\mathbf{x},\mathbf{y}))}{\sum_{\mathbf{x}'\in\mathcal{X}} \exp(q_{\boldsymbol{\theta}_c}(\mathbf{x}',\mathbf{y}))}, \tag{27}$$

where $q_{\boldsymbol{\theta}_c}(\mathbf{x},\mathbf{y}) \;=\; \log P_{\boldsymbol{\theta}_c}(\mathbf{y}\,|\,\mathbf{x}) \;+\; \log p(\mathbf{x})$ is defined for convenience. Let us introduce a temperature constant $\tau > 0$ in the softmax function, and define the temperature-scaled posterior distribution

$$P_{\boldsymbol{\theta}_c}^{(\tau)}(\mathbf{x}\,|\,\mathbf{y}) \;=\; \frac{\exp(q_{\boldsymbol{\theta}_c}(\mathbf{x},\mathbf{y})\,/\,\tau)}{\sum_{\mathbf{x}'\in\mathcal{X}} \exp(q_{\boldsymbol{\theta}_c}(\mathbf{x}',\mathbf{y})\,/\,\tau)}. \tag{28}$$

For large $\tau$, the above posterior approaches a uniform distribution. For small $\tau$ it approaches a distribution with probability $1$ for $\mathbf{x}$ corresponding to the maximum exponent, and $0$s elsewhere. Based on this observation, we define the following smooth approximation of the MAP SE layer

$$\widehat{\mathbf{X}}_{\mathrm{soft}}(\mathbf{y}) \;=\; \sum_{\mathbf{x}\in\mathcal{X}} P_{\boldsymbol{\theta}_c}^{(\tau)}(\mathbf{x}\,|\,\mathbf{y})\,\mathbf{x} \tag{29}$$

This can be interpreted as the conditional expectation of $\mathbf{x}$ given $\mathbf{y}$ with respect to the temperature-scaled posterior distribution (28). We can show that this smooth MAP estimate approaches the true MAP estimate in the limit as $\tau$ approaches 0, *i.e.*,

$$\lim_{\tau \to 0} \sum_{\mathbf{x} \in \mathcal{X}} P_{\boldsymbol{\theta}_c}^{(\tau)}(\mathbf{x} \,|\, \mathbf{y}) \, \mathbf{x} \; = \; \operatorname*{argmax}_{\mathbf{x} \in \mathcal{X}} P_{\boldsymbol{\theta}_c}(\mathbf{x} \,|\, \mathbf{y}).$$

**Training Based on Temperature Annealing.**

The MAP symbol estimation autoencoder uses the smooth MAP estimate (29) during training, and the exact MAP estimate during inference. In order to have good convergence and to prevent the training from getting stuck at poor solutions, we *do not fix* the temperature $\tau$ to a small value throughout. Instead, we decrease $\tau$ according to a temperature annealing schedule during training. Specifically, $\tau$ is initialized to a reasonably large value (*e.g.*, $\tau_i = 1$), and it is decreased by an exponential factor $\eta \in (0, 1)$ at the end of every $r \geq 1$ epochs. The solution (autoencoder parameters) at the end of $r$ epochs for the current temperature is used to initialize the training at the next lower temperature. This process is continued until a small final temperature $\tau_f$ is reached. In our experiments, we set the constants related to temperature annealing as follows: $\tau_i = 1, \tau_f = 0.05, r = 10, \eta = 0.7169$. This choice of $\eta$ ensures that there are 10 temperature steps including the initial and final values.

# D  ADDITIONAL EXPERIMENTS

## D.1  EXPERIMENTAL SETUP

We implemented the mixture density network and communication autoencoder models using TensorFlow (Abadi et al., 2015) and TensorFlow Probability. We used the BFGS optimizer implementation available in TensorFlow Probability. The code base for our work can be found at https://anonymous.4open.science/r/domain_adaptation-7C0D/. All the experiments were run on a Macbook Pro laptop with 16 GB memory and 8 CPU cores. Table 4 summarizes the architecture of the encoder, MDN (channel model), and decoder neural networks. Note that the output layer of the MDN is a concatenation (denoted by $\oplus$) of three fully-connected layers predicting the means, variances, and mixing prior logit parameters of the Gaussian mixture. The size of the hidden layers $n_h$ was set to 100.

Table 4: Architecture of the Encoder, MDN channel, and Decoder neural networks. FC - fully connected (dense) layer; $\oplus$ denotes layer concatenation; ELU - exponential linear unit; $m$ - number of messages; $d$ - encoding dimension; $k$ - number of mixture components; $n_h$ - size of a hidden layer.

| Network | Layer | Activation |
|---|---|---|
| Encoder | FC, $m \times n_h$ | ReLU |
| | FC, $n_h \times d$ | Linear |
| | Normalization (avg. power) | None |
| MDN | FC, $d \times n_h$ | ReLU |
| | FC, $n_h \times n_h$ | ReLU |
| | FC, $n_h \times kd$ (means) | Linear |
| | $\oplus$ FC, $n_h \times kd$ (variances) | ELU $+ 1 + \epsilon$ |
| | $\oplus$ FC, $n_h \times k$ (prior logits) | Linear |
| Decoder | FC, $d \times n_h$ | ReLU |
| | FC, $n_h \times m$ | Softmax |

The parameters $\psi$ of the proposed adaptation method are initialized as follows for each component $i$:

$$\mathbf{A}_i = \mathbf{I}_d, \;\; \mathbf{b}_i = \mathbf{0}, \;\; \mathbf{C}_i = \mathbf{I}_d, \;\; \beta_i = \mathbf{1}, \;\; \gamma_i = \mathbf{0},$$

where $\mathbf{I}_d$ is the $d \times d$ identity matrix. This initialization ensures that the target Gaussian mixture is always initialized with the source Gaussian mixture. The regularization constant $\lambda$ in the adaptation objective was varied over 16 equally-spaced values on the log-scale (base 10) with range $10^{-5}$ to 100. The $\lambda$ value and $\psi$ corresponding to the smallest validation metric are selected as the final solution.

We used the Adam optimizer (Kingma & Ba, 2015) with a fixed learning rate of 0.001, batch size of 128, and 100 epochs for training the MDN. For adaptation of the MDN using the baseline methods transfer and transfer-last-layer, Adam is used with the same learning rate for 200 epochs. The batch size is set as $b = \max\{10, 0.1 \, N_c^{(t)}\}$, where $N_c^{(t)}$ is number of adaptation samples in the target domain. For training the autoencoder using Algorithm 1, we found that stochastic gradient descent (SGD) with Nesterov momentum (constant 0.9), and an exponential learning rate schedule between 0.1 and 0.005 works better than Adam.

### D.2 DETAILS ON THE FPGA EXPERIMENT

Referring to the experiment in § 5.3, for the real and over-the-air traces we used the platform from Lacruz et al. (2021). This ultra-wide-band mm-wave transceiver baseband memory-based design is developed on top of an ZCU111 RFSoC FPGA. This evaluation board features a Zynq Ultrascale + ZCU28DR. This FPGA is equipped with $8 \times 8$ AD/DA converters with giga-sampling capabilities, which make it ideal for RF system development; the 4 GB DDR4 memories contain RF-ADCs with up to 4 GSPS of sampling rate, and RF-DACs with up to 6.544 GSPS. This board also includes a quad-core ARM Cortex-A53 and a dual-core ARM Cortex-R5 real-time processor.

For the radio frequency, we used 60 GHz RF front-end antennas. These kits include a $16 + 16$ TRX patch array antenna plus the RF module with up/down conversion from baseband to I/Q channels, and TX/RX local oscillator (LO) frequency control. The antennas use $57 - 71$ GHz, a range of frequencies that cover the unlicensed 60 GHz band for mm-wave channels, and are managed from a PC Host via USB.

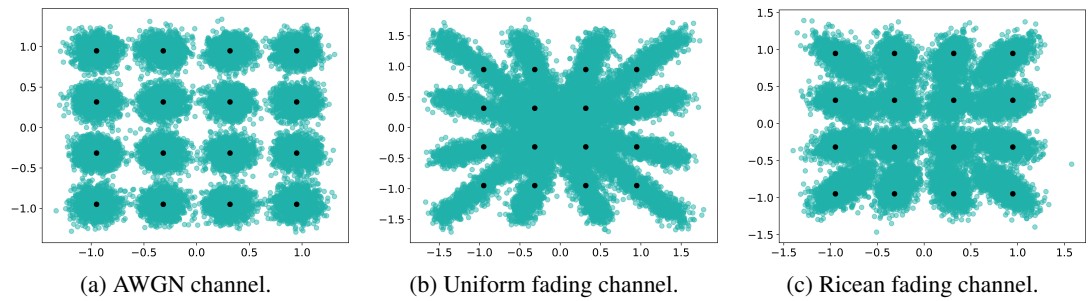

(a) AWGN channel.    (b) Uniform fading channel.    (c) Ricean fading channel.

Figure 8: Plot of different simulated channel models with a 16-QAM constellation as the channel input for an SNR of 14 dB.

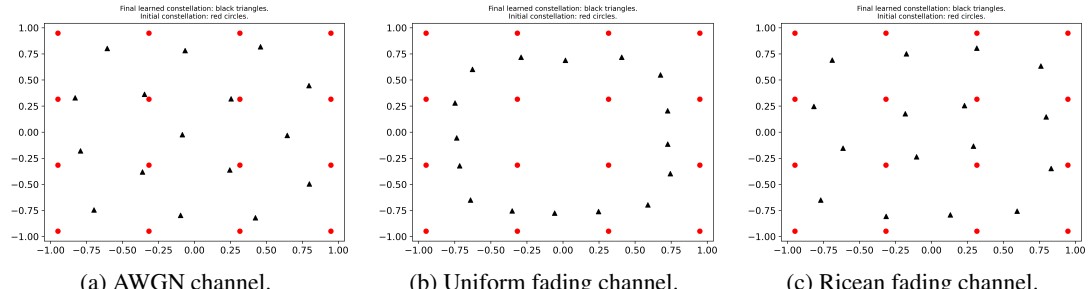

(a) AWGN channel.    (b) Uniform fading channel.    (c) Ricean fading channel.

Figure 9: The symbol constellation learned by the autoencoder on different channel distributions. Starting from the 16-QAM constellation (red circles), the autoencoder learns a custom constellation (black triangles).

We implemented a hardware on the loop training. For the experimentation on real traces, we use Matlab as a central axis. The PC host running Matlab is connected to the platform via Ethernet. The FPGA can transmit different custom waveforms like 16-QAM frames from the 802.11ad and 802.11ay standards, with 2 GHz of bandwidth. The frames are sent over-the-air via 60 GHz radio frequency kits, and the samples are stored at the FPGA DDR memory. We decode the received data from the transmission, removing the preamble and header fields and extracting the symbols to train the MDN. We add a preamble to the generated constellation from the MDN for packet detection purposes, and we transmit again the new waveforms over-the-air. Finally, the adaptation is performed offline with the decoded symbols from the custom autoencoder-learned constellation.

**Source and Target Domains.**

For the experiment in § 5.3, we introduced distribution changes via IQ imbalance based distortions to the symbol constellation, and evaluated the adaptation performance as a function of the level of imbalance. The source domain would be the original channel, the over-the-air link between the transmitter and receiver on which the training data is collected. This source domain data is used

for training the channel and the autoencoder. The target domain would be a modification of the source domain where the symbols used by the transmitter (e.g., 16-QAM) are distorted by modifying the in-phase and quadrature-phase (IQ) components of the RF signal. This causes a change in the distribution observed by the receiver (decoder), leading to a drop in performance without any adaptation.

### D.3 COMPARISON WITH A BENCHMARK RETRAINED AUTOENCODER

For the experiment setting of § 5.2, we include an additional benchmark comparison with a fully-retrained autoencoder. The retrained autoencoder's performance would be the best-case achievable performance by any adaptation method, given that retraining uses plenty of labeled training data and trains for much longer. Note that when we say the autoencoder is retrained using Algorithm 1, it is implied that the MDN channel model is also retrained.

Recall that for the adaptation methods, the autoencoder is *trained once* using channel data from a source distribution A, at an SNR of $14$ dB. The adaptation methods use a small target dataset from a target distribution B, whose SNR is varied from 8 dB to 20 dB. The following protocol is used for retraining the autoencoder. Suppose we are adapting from a source distribution A to a target distribution B, the autoencoder is retrained only on data from the target distribution B *for each SNR* from 8 dB to 20 dB. We used $25,000$ samples for training the MDN channel and $300,000$ samples for training the autoencoder. The MDN channel is optimized for 100 epochs during a single outer epoch of optimizing the autoencoder (encoder and decoder networks). The iterative training of the autoencoder and MDN channel is repeated for 20 outer epochs (which we found to be sufficient for convergence).

The performance comparison with a retrained autoencoder for the proposed adaptation methods i) affine transformation and ii) MAP symbol estimation are given in Fig. 10 and Fig. 11 respectively. Note that the plots for the proposed and baseline methods are exactly the same as in Fig. 4 and Fig. 5. Only the retrained autoencoder (labeled "Retrained autoenc") has been added to the plots. As one might expect, the retrained autoencoder has a significantly lower BLER than the other methods. In scenarios such as adaptation from AWGN to Uniform fading, the distribution change can be so large (see Fig. 8) that it may not be sufficient to just adapt the channel model in order to close the gap. It may be necessary to also optimize the encoder's constellation and the decoder for the target channel distribution.

Figure 9 shows the optimal constellation learned by the autoencoder (in black) for different channel distributions at 14 dB SNR. The constellation for Uniform fading resembles an M-PSK (phase-shift keying) modulation, while the constellation for AWGN is very different. This implies that the optimal constellation for an AWGN channel is very different from that of a Uniform fading channel.

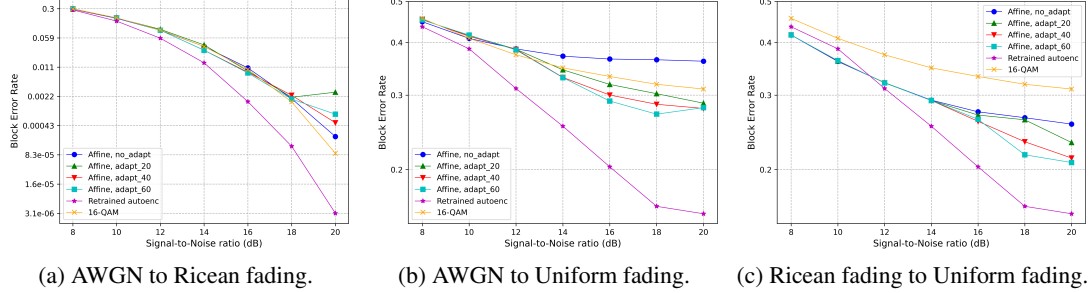

(a) AWGN to Ricean fading.        (b) AWGN to Uniform fading.        (c) Ricean fading to Uniform fading.

Figure 10: Comparison of the affine transformation based adaptation to a fully-retrained autoencoder.

### D.4 COMPARISON BETWEEN THE PROPOSED DECODER ADAPTATION METHODS

In this experiment we compare the performance of the proposed decoder adaptation methods in § 4.1 and § 4.2 on simulated channel variations. Figures 12, 13, and 14 compare the performance of the two methods on different source/target channels using 20, 40, and 60 samples per symbol for adaptation. We observe that in most cases, the MAP SE method outperforms the affine transformation based method. This could be due to the fact that MAP SE has an optimal transformation from the

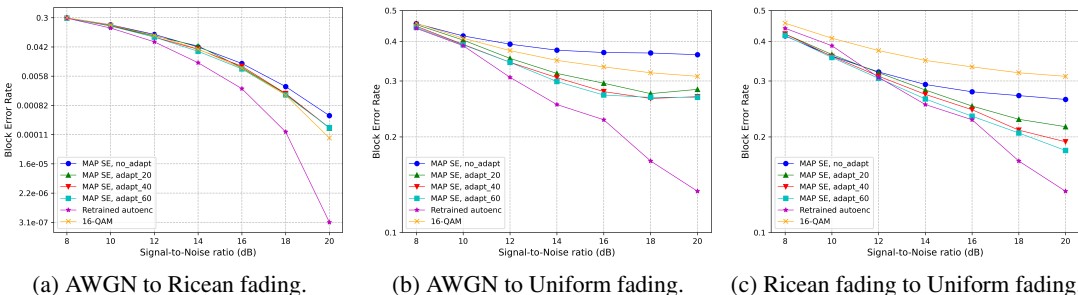

(a) AWGN to Ricean fading.  (b) AWGN to Uniform fading.  (c) Ricean fading to Uniform fading.

Figure 11: Comparison of the MAP SE based adaptation to a fully-retrained autoencoder.

channel output **y** to the most probable input **x**, which better compensates for changes in the channel distribution.

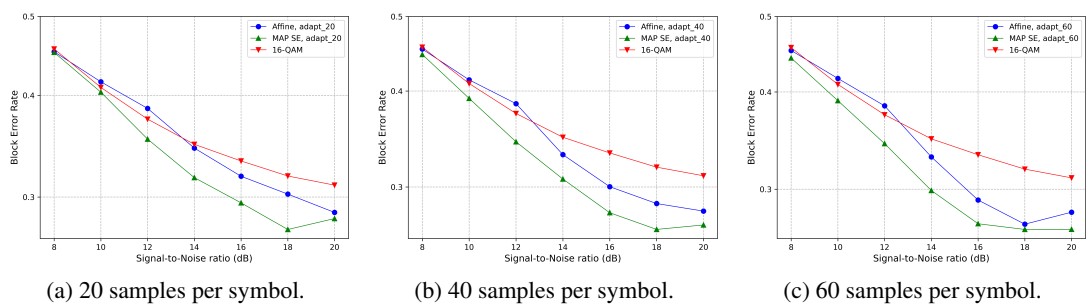

(a) 20 samples per symbol.  (b) 40 samples per symbol.  (c) 60 samples per symbol.

Figure 12: Comparison of the affine transformation and MAP SE adaptation methods when the source channel is **AWGN** and the target channel is **Uniform fading**. 16-QAM is included simply as a reference.

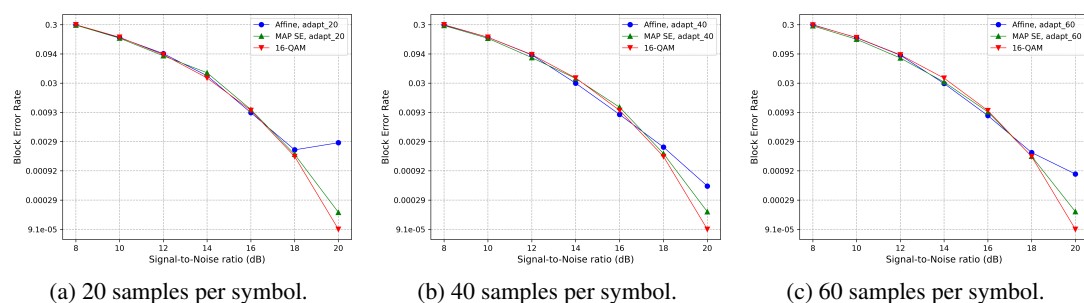

(a) 20 samples per symbol.  (b) 40 samples per symbol.  (c) 60 samples per symbol.

Figure 13: Comparison of the affine transformation and MAP SE adaptation methods when the source channel is **AWGN** and the target channel is **Ricean fading**.

### D.5 PERFORMANCE UNDER NO DISTRIBUTION CHANGE

We studied the performance of the MDN adaptation method when the source and target domains are the same, and compared with the two baselines using 50 randomly generated Gaussian mixture datasets. Figure 15 illustrates one such randomly generated training, adaptation, and test data set (the source and target distribution are not the same in this figure). The results in terms of conditional log-likelihood are shown in Table 5 (second line). Compared to the results when the source and target domains are different, we find that our method has a significant better performance compared to the baselines when the source and target domains. This demonstrates that the proposed MDN adaptation is capable of handling well scenarios where the distribution does not change, or changes minimally.

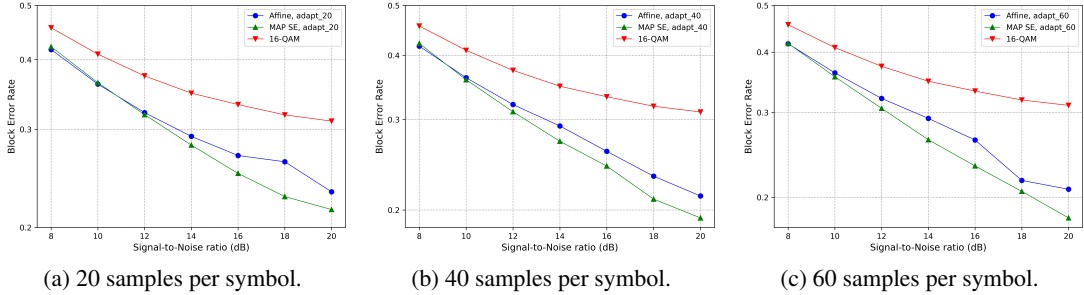

(a) 20 samples per symbol.  (b) 40 samples per symbol.  (c) 60 samples per symbol.

Figure 14: Comparison of the affine transformation and MAP SE adaptation methods when the source channel is **Ricean fading** and the target channel is **Uniform fading**.

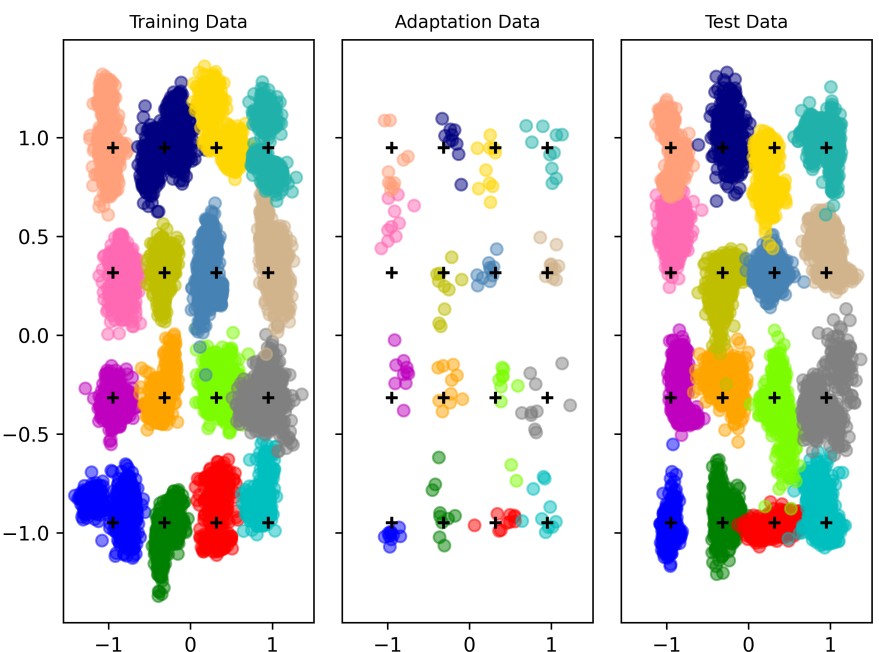

Figure 15: Training, adaptation, and test dataset generated from different random Gaussian mixtures for the source and target domains.

### D.6 PERFORMANCE WHEN THE NUMBER OF COMPONENTS ARE MISMATCHED

We also studied how much impact is introduced if the number of components is mismatched in the ground truth source and target Gaussian mixtures. Using 50 randomly-generated Gaussian mixture datasets, these results are summarized in Table 5 (third line). Comparing the results with that of the case when the number of components are matched, we find that although the performance is slightly worse than the case when the number of components are matched, the performance drop is not significant.

### D.7 SELECTION OF THE HYPER-PARAMETER $\lambda$

We investigated the sensitivity of the proposed automatic selection of $\lambda$ (using the validation metric) by comparing the performance of automatic selection with that of using different fixed $\lambda$ values. These results are summarized in Table 6. Note that the automatic selection searches over 16 log-spaced $\lambda$ values ranging from 1e-5 to 100. From the table, we observe that the conditional log-likelihood corresponding to automatic selection of $\lambda$ is sometimes worse than that of other fixed values of $\lambda$. However, the results are better than those when $\lambda$ is 0 (except for Ricean fading to uniform fading). Therefore, by setting a non-zero $\lambda$, the proposed adaptation method increases the final conditional

Table 5: Conditional log-likelihood of the MDN adaptation methods on test data from the target domain. The source and target domain datasets are generated from random, class-conditional Gaussian mixtures. **Same GMM** refers to the special case where there is no distribution change. For the case #**components mismatched**, the number of components in the source and target Gaussian mixtures is different.

| Source / Target distribution | #Target samples | Proposed | | Transfer | | Transfer-last-layer | |
|---|---|---|---|---|---|---|---|
| | | median | 95% CI | median | 95% CI | median | 95% CI |
| Different GMMs | 80 | **0.78** | (-0.44, 1.07) | -2.45 | (-5.83, -0.49) | -2.87 | (-7.14, -0.52) |
| | 160 | **1.16** | (0.86, 1.37) | 0.80 | (-0.01, 1.34) | 0.66 | (-0.28, 1.06) |
| | 240 | 1.30 | (1.05, 1.51) | **1.41** | (0.98, 1.63) | 1.32 | (1.01, 1.58) |
| | 320 | 1.44 | (1.15, 1.68) | **1.63** | (1.31, 1.79) | 1.57 | (1.33, 1.76) |
| Same GMM | 160 | **2.11** | (1.94, 2.31) | 0.91 | (0.06, 1.39) | 0.54 | (-0.53, 1.26) |
| Different GMMs, #component mismatched | 160 | **0.99** | (-0.83, 1.36) | 0.61 | (-0.58, 1.18) | 0.40 | (-0.83, 1.04) |

Table 6: Conditional log-likelihood of the proposed MDN adaptation for different fixed values of the hyper-parameter $\lambda$. This is compared with the automatic selection of $\lambda$ based on the validation metric.

| Source channel | Target channel | $\lambda$ values | Proposed | |
|---|---|---|---|---|
| | | | median | 95% CI |
| AWGN | Uniform fading | 0 | -0.47 | (-2.21, 0.48) |
| | | 1e-5 | -0.87 | (-2.26, -0.47) |
| | | 1e-4 | -0.06 | (-1.30, -0.56) |
| | | 1e-3 | -0.29 | (-1.90, 0.64) |
| | | 1e-2 | 0.17 | (-0.79, 0.61) |
| | | 1e-1 | -0.12 | (-1.37, 0.59) |
| | | 1 | -0.74 | (-1.97, 0.03) |
| | | auto | -0.29 | (-1.97, 0.58) |
| AWGN | Ricean fading | 0 | 1.21 | (1.02, 1.38) |
| | | 1e-5 | 1.23 | (1.03, 1.37) |
| | | 1e-4 | 1.21 | (1.02, 1.37) |
| | | 1e-3 | 1.26 | (1.01, 1.38) |
| | | 1e-2 | 1.25 | (1.10, 1.35) |
| | | 1e-1 | 1.30 | (-1.06, 1.38) |
| | | 1 | 0.87 | (-0.18, 1.29) |
| | | auto | 1.24 | (0.80, 1.38) |
| Ricean fading | Uniform fading | 0 | -0.14 | (-1.89, 0.64) |
| | | 1e-5 | -0.14 | (-1.86, 0.61) |
| | | 1e-4 | -0.30 | (-1.52, 0.66) |
| | | 1e-3 | -0.12 | (-1.36, 0.62) |
| | | 1e-2 | 0.12 | (-1.08, 0.65) |
| | | 1e-1 | -0.11 | (-0.87, 0.70) |
| | | 1 | -0.54 | (-1.43, 0.53) |
| | | auto | -0.41 | (-4.24, 0.70) |

log-likelihood in most cases. A more extensive study exploring some alternate methods for selecting $\lambda$ would be beneficial.

# E  SIMULATED CHANNEL VARIATION MODELS

We provide details of the mathematical models used to create simulated channel variations in our experiments. These models are frequently used in the study of wireless channels (Goldsmith, 2005).

### E.1 UNIFORM FADING MODEL

The channel output $\mathbf{y} \in \mathbb{R}^d$ for this model as a function of the channel input (modulated symbol vector) $\mathbf{x} \in \mathbb{R}^d$ is given by

$$\mathbf{y} = A\,\mathbf{x} + \mathbf{n},$$

where $A \sim \mathrm{Unif}[0, a]$ is a uniformly-distributed scale factor, and $\mathbf{n} \sim \mathcal{N}(\cdot \,|\, \mathbf{0}, \sigma_0^2\, \mathbf{I}_d)$ is an additive Gaussian noise vector. Both $A$ and $\mathbf{n}$ are assumed to be independent of each other and $\mathbf{x}$. The average power in the signal component of $\mathbf{y}$ is given by

$$\widetilde{p}_{\mathrm{avg}} := \mathbb{E}[\|A\,\mathbf{x}\|_2^2] = \mathbb{E}[A^2]\,\mathbb{E}[\|\mathbf{x}\|_2^2]$$
$$= \frac{a^2}{3}\, p_{\mathrm{avg}},$$

where $p_{\mathrm{avg}}$ denotes the average power in the channel input $\mathbf{x}$. The noise power in this case is given by $\mathbb{E}[\|\mathbf{n}\|_2^2] = \sigma_0^2$. The signal-to-noise ratio (SNR) for this model is therefore given by

$$\frac{E_b}{N_0} = \frac{\mathbb{E}[\|A\,\mathbf{x}\|_2^2]}{2\,R\,\mathbb{E}[\|\mathbf{n}\|_2^2]} = \frac{a^2\, p_{\mathrm{avg}}}{6\,R\,\sigma_0^2},$$

where $R$ is the communication rate of the system in bits/channel use. We select the fading factor $a$ such that the channel output has a target SNR value using the following equation:

$$a = \sqrt{\frac{6\,R\,\sigma_0^2\,(E_b/N_0)}{p_{\mathrm{avg}}}}. \tag{30}$$

### E.2 RICEAN AND RAYLEIGH FADING MODELS

The channel output for the Ricean fading model is given by

$$\mathbf{y} = \mathbf{A}\,\mathbf{x} + \mathbf{n},$$

where $\mathbf{A}$ is a diagonal matrix with the diagonal elements $a_1, \cdots, a_d \overset{\mathrm{iid}}{\sim} \mathrm{Rice}(\cdot \,|\, \nu, \sigma_a^2)$ following a Rice distribution, and $\mathbf{n} \sim \mathcal{N}(\cdot \,|\, \mathbf{0}, \sigma_0^2\, \mathbf{I}_d)$ is an additive Gaussian noise vector. It is assumed that $\mathbf{n}$ and $\mathbf{A}$ are independent of each other and of $\mathbf{x}$. Note that Rayleigh fading is a special case of Ricean fading when the parameter $\nu = 0$. For this model, the average power in the signal component of $\mathbf{y}$ is given by

$$\widetilde{p}_{\mathrm{avg}} := \mathbb{E}[\|\mathbf{A}\,\mathbf{x}\|_2^2] = \sum_{i=1}^{d} \mathbb{E}[a_i^2\, x_i^2] = \sum_{i=1}^{d} \mathbb{E}[a_i^2]\,\mathbb{E}[x_i^2]$$
$$= (2\,\sigma_a^2 + \nu^2)\,\mathbb{E}[\|\mathbf{x}\|_2^2] = (2\,\sigma_a^2 + \nu^2)\, p_{\mathrm{avg}},$$

where $p_{\mathrm{avg}}$ denotes the average power in the channel input $\mathbf{x}$. We used the fact that the second moment of the Rice distribution is given by $\mathbb{E}[a_i^2] = 2\,\sigma_a^2 + \nu^2$. It is useful to consider the derived parameters $K = \nu^2 / 2\,\sigma_a^2$ which corresponds to the ratio of power along the line-of-sight (LoS) path to the power along the remaining paths, and $\Omega = 2\,\sigma_a^2 + \nu^2$ which corresponds to the total power received along all the paths. The SNR for this model is given by

$$\frac{E_b}{N_0} = \frac{\mathbb{E}[\|\mathbf{A}\,\mathbf{x}\|_2^2]}{2\,R\,\mathbb{E}[\|\mathbf{n}\|_2^2]} = \frac{(2\,\sigma_a^2 + \nu^2)\, p_{\mathrm{avg}}}{2\,R\,\sigma_0^2}.$$

For a given input average power and target SNR, the parameters of the Rice distribution can be set using the equation

$$2\,\sigma_a^2 + \nu^2 = \frac{2\,R\,\sigma_0^2\,(E_b/N_0)}{p_{\mathrm{avg}}}.$$

To create channel variations of different SNR, we fix the variance $\sigma_a^2$ and vary the power of the LoS component $\nu^2$. Suppose the smallest SNR value considered is $S_{\mathrm{min}}$, we set $\sigma_a^2$ using

$$2\,\sigma_a^2 = \frac{2\,R\,\sigma_0^2\,S_{\mathrm{min}}}{p_{\mathrm{avg}}}, \tag{31}$$

and set $\nu$ to achieve a target SNR $E_b/N_0$ using

$$\nu^2 = \frac{2\,R\,\sigma_0^2\,(E_b/N_0 - S_{\min})}{p_{\mathrm{avg}}}. \tag{32}$$

For this choice of parameters, the power ratio of LoS to non-LoS components is given by

$$K = \frac{E_b\,/\,N_0}{S_{\min}} - 1.$$

The $K$-factor for Rician fading in indoor channel environments with an unobstructed line-of-sight is typically in the range $4\,\mathrm{dB}$ to $12\,\mathrm{dB}$ (Linnartz, 2001). Rayleigh fading is obtained for $K = 0$ (or $\nu = 0$).

Finally, note that the vector $\mathbf{x}$ is composed of one or more pairs of in-phase and quadrature (IQ) components of the encoded signal (dimension can be expressed as $d = 2m$). Since each IQ component is transmitted as a single RF signal, the Ricean amplitude scale is kept the same for successive pairs of IQ components in $\mathbf{x}$. In other words, the amplitude scales are chosen to be $a_1, a_1, \cdots, a_m, a_m$. This does not change any of the above results.

