# OpenReview forum: "Fast and Sample-Efficient Domain Adaptation for Autoencoder-Based End-to-End Communication"
_ICLR.cc/2022/Conference — ICLR 2022 Submitted_

### Official Review · Reviewer_S81w · 2021-10-29

**Correctness:** 3
**Technical Novelty And Significance:** 2
**Empirical Novelty And Significance:** 2
**Recommendation:** 5
**Confidence:** 4

**Main Review:**

The addressed problem is of practical relevance and the authors carry out experiments on both real and simulated data, with good results.
However, I have some reservations.

What is the focus of the paper? Is it the domain adaptation approach or the specific problem of domain adaptation (DA) of autoencoders using an MDN as the channel model. I am asking because from the abstract it seems that the main focus is on the adaptation technique and the autoencoder case is just an application examples, while from the introduction section it seems that the main focus is on the autoencoder.

The authors assume that the source and target distribution have the same number of components with a one-to-one correspondence. How realistic is the working hypothesis? In the appendix the authors report the results if the number of components is mis-matched. I think it should be included in the main body of the paper instead.
Also, how realistic is the working hypothesis that labelled samples from the target domain an be collected before the channel changes again? Wireless channels can vary very rapidly, so I presume that the proposed approach applies only relatively stable channels.

It would be interesting to include cases in which the domain adaptation fails. I presume that if the channel changes significantly then the proposed adaptation might fail.

In (section 4.1), the authors adopt the MAP rule to determine the channel input and the Gaussian component. How complex is the MAP rule? I presume that if the number of Gaussian components and the constellation size become large, it might be an issue. (The same question applies to section 4.2 in which the MAP rule is applied as well). I am not sure if in Appendix C.4 the authors included the complexity of MAP rule. I think they should as it is part of the overall approach.

I am confused about the way results are reported in Table 1. The authors report the estimated conditional likelihood p(x/y) estimate by the various methods, but what is the true p(x/y)? How to evaluate if the estimated conditional likelihood is close to the true one? Also, it seems that for large number of samples the “Transfer-last-layer” outperforms the proposed approach. Why is this the case?

Why in Fig 4 and 5 the 16 QAM performs less than the no adaptation?

Under “observations and takeaways” the authors state that “Between the two adaptation methods, MAP SE performs marginally better than the Affine method.” Which one of the two would the author recommend? Are they equivalent in terms of complexity?

Since the authors claim that the proposed approach is fast and light-weight, the authors should include a study of the complexity analysis in the main body of the paper (I suggest move it there from the appendix), including also the training and running times of the proposed approach. I see that they include the number of trainable parameters, which is very important, but it’s only partial information.

The baselines included in the numerical results are very similar to the proposed approach. It would be interesting if the authors could include other methods as well, maybe even more complex, to analyze the trade-off between performance and complexity offered by  the proposed approach.

Maybe the authors could include a future work section in the conclusion (if they assume to continue research on this topic) and discuss the generality of this approach. What are other possible application areas?


**Summary Of The Paper:**

This paper addressed the problem of domain adaptation when the target domain has only few labeled samples. It proposes to use an MDN with Gaussian components to represent the source distribution and then adapt the MDN parameters to match the target distribution to the source distribution.
This approach is adopted to address the problem of adapting a communication autoencoder (trained on the source distribution) to samples drawn from the target distribution without worsening the error rate of the autoencoder. The paper includes results on simulated and real data.


**Summary Of The Review:**

The addressed problem is of interest and the authors test the proposed technique both on simulate and real data. However, the main technical contribution of the paper relies on the well-known property of Gaussian distributions (hence the paper provides only marginal novelty) and the two working hypothesis (same number of components between the two Gaussian distributions and a one-to-one
correspondence between the components) are not properly discussed from a practical viewpoint. Finally, considering the fact that the authors promote their approach because it is fast and light-weight, a proper complexity analysis is missing in the paper (the one in the appendix is partial and does not seem to include the cost of the MAP rule).

---

> ### Author Response · Authors · 2021-11-23
> **Author response to reviewer S81w (part 1)**
>
> We thank the reviewer for their comments and feedback on the paper. They have been very helpful in revising and improving our submission. Please find our responses to your questions and comments below.
>
> > What is the focus of the paper? Is it the domain adaptation approach or the specific problem of domain adaptation (DA) of autoencoders using an MDN as the channel model. I am asking because from the abstract it seems that the main focus is on the adaptation technique and the autoencoder case is just an application examples, while from the introduction section it seems that the main focus is on the autoencoder.
>
> The main focus of our paper is on the domain adaptation of an end-to-end autoencoder-based communication system, under a frequently-changing channel distribution. We also focus on a generative channel model based on mixture density networks. We agree with the reviewer that our earlier abstract seemed to focus more on the general domain adaptation problem. We have now revised the abstract to focus more on the specific problem of autoencoder and channel adaptation that we address in the paper.
>
> > The authors assume that the source and target distribution have the same number of components with a one-to-one correspondence. How realistic is the working hypothesis? In the appendix the authors report the results if the number of components is mis-matched. I think it should be included in the main body of the paper instead.
>
> In the revised paper, we state these two assumptions and provide a brief justification for the same. The assumption of same number of components is a practical one that we make in order to not have to change the architecture of the MDN. Adding or removing components would require a change to the output layer of the MDN. Also, this assumption can be practically justified when the number of components is chosen to be sufficiently large. With a large number of components, the appearance or disappearance of a few clusters of data from the true data distribution can be compensated for by other components in the Gaussian mixture.
>
> The assumption of one-to-one correspondence between the components makes it mathematically convenient to derive a closed-form expression for the KL-divergence between two Gaussian mixtures, which would not be possible in the general case. We believe that both the assumptions are reasonably realistic when the change in distribution is not very drastic. As pointed out by the reviewer, we have performed an ablation study to test the scenario of component mismatch using random synthetic Gaussian mixtures (Appendix D.6). Due to the page limit, we had to prioritize certain experiments to be included in the main paper, and move some of the ablation experiments to the Appendix.
>
> > Also, how realistic is the working hypothesis that labelled samples from the target domain an be collected before the channel changes again? Wireless channels can vary very rapidly, so I presume that the proposed approach applies only relatively stable channels.
>
> The hypothesis is that a small number of labelled samples can be collected for adaptation under changing channel conditions. This is a realistic hypothesis for the wireless setting because a small number of training samples can be collected from the preamble always included in the packet headers. This does not usually require the transmission of additional non-data packets. The labeling part is easy in this problem setting because the type of symbol transmitted has a one-to-one mapping to the input message. Therefore, the label is obtained for free.
>
> > It would be interesting to include cases in which the domain adaptation fails. I presume that if the channel changes significantly then the proposed adaptation might fail.
>
> We agree that it would be interesting to evaluate on scenarios where the proposed domain adaptation can fail. Some of the distribution changes on simulated channels, such as AWGN to Uniform fading, are actually significant and increase the block error rate of the autoencoder (especially when the SNR is also different). See for example, the distribution plots for AWGN and Uniform fading channels in Figure 8 in Appendix D. From Figures 10(b) and 11(b) in Appendix D.3, we observe that the improvement in block error rate of the proposed method is good compared to no adaptation, but is still much higher than the benchmark retrained autoencoder. For such large distribution changes, it may be necessary to allow the encoder and decoder networks to adapt as well (the learned constellation may need to be updated).

---

> > ### Author Response · Authors · 2021-11-23
> > **Author response to reviewer S81w (part 2)**
> >
> > > In (section 4.1), the authors adopt the MAP rule to determine the channel input and the Gaussian component. How complex is the MAP rule? I presume that if the number of Gaussian components and the constellation size become large, it might be an issue. (The same question applies to section 4.2 in which the MAP rule is applied as well). I am not sure if in Appendix C.4 the authors included the complexity of MAP rule. I think they should as it is part of the overall approach.
> >
> > We have revised the computational complexity discussion in Appendix C.4 to discuss both adaptation methods proposed in section 4. The complexity of the affine transformation method (section 4.1) can be expressed as $O(k^2 m^2 d^3)$ and that of the MAP SE method is $O(k m^2 d^3)$. Here $m$ is the number of symbols, $k$ is the number of components, and $d$ is the dimension. The reviewer is correct in observing that when $m$ and/or $k$ is large, the complexity of adaptation would be high. It has also been observed in the literature that the computational and memory complexity of end-to-end learning of a symbol-level autoencoder grows quickly for large $m$, which is an open problem.
> >
> > > I am confused about the way results are reported in Table 1. The authors report the estimated conditional likelihood p(x/y) estimate by the various methods, but what is the true p(x/y)? How to evaluate if the estimated conditional likelihood is close to the true one?
> >
> > In Table 1, we have reported the conditional log-likelihood of the adapted MDN using different methods, which allows for a relative comparison of the methods. Since these are synthetic channels, we agree with the reviewer that it is possible to compute the true data log-likelihood under the target channel (e.g. Ricean fading). Unfortunately, we were not able to collect this metric to include in the revised paper.
> >
> > > Also, it seems that for large number of samples the “Transfer-last-layer” outperforms the proposed approach. Why is this the case?
> >
> > The “transfer-last-layer” method does have a slightly higher conditional log-likelihood in some cases when the number of adaptation samples is relatively large. We do not have a concrete reason for this, but we suspect that the use of Adam optimizer with many epochs (200 epochs was used for “transfer-last-layer”) is beneficial when the number of adaptation samples is large. The stochastic nature of mini-batch SGD updates could be helping, whereas the proposed method uses a deterministic BFGS based optimization.
> >
> > > Why in Fig 4 and 5 the 16 QAM performs less than the no adaptation?
> >
> > The reviewer is probably referring to the case of adaptation from AWGN to Ricean fading. In this case, 16-QAM has a lower BLER than the proposed methods for the high SNR range. This could be due to the fact that the Ricean fading channel output becomes well-separated (between the symbols) at high SNR. We have data plots to verify this observation. As a result, even the simple 16-QAM decoder is able to have very good decoding performance.
> >
> > > Under “observations and takeaways” the authors state that “Between the two adaptation methods, MAP SE performs marginally better than the Affine method.” Which one of the two would the author recommend? Are they equivalent in terms of complexity?
> >
> > In the revised paper, we include a direct comparison of the performance of the two methods in Appendix D.4 (Figs. 12, 13, and 14). The improvement in BLER of the MAP SE is clear from these results. From the discussion in Appendix C.4, both the methods have a comparable computational complexity, but the affine transformation based method has a higher complexity by a factor of $k$ (the number of components). Therefore, I would recommend the MAP SE method.

---

> > > ### Author Response · Authors · 2021-11-23
> > > **Author response to reviewer S81w (part 3)**
> > >
> > > > Since the authors claim that the proposed approach is fast and light-weight, the authors should include a study of the complexity analysis in the main body of the paper (I suggest move it there from the appendix), including also the training and running times of the proposed approach. I see that they include the number of trainable parameters, which is very important, but it’s only partial information.
> > >
> > > We agree that complexity analysis is an important aspect of the proposed  method. In the revised paper, we have provided more details on the computational complexity in Appendix C.4. Unfortunately, we were not able to find sufficient space to include this discussion in the main paper.
> > >
> > > On the comparison of running times: our implementation of the proposed method is fully in TensorFlow and includes many custom modules that were hard to fully optimize in order to run at speeds comparable to the core TensorFlow algorithms, that are graph-compiled. On the other hand, the baseline adaptation methods such as “Transfer” and “Transfer-last-layer” do *not* require any custom implementation, and therefore run at graph-compiled speeds. For this reason, we have not directly included a comparison of the running times, since it could be misleading. We would however add that the proposed method runs much faster compared to retraining, and is a factor of 3 to 4x slower than the baselines (based on our non-optimized implementation).
> > >
> > > > The baselines included in the numerical results are very similar to the proposed approach. It would be interesting if the authors could include other methods as well, maybe even more complex, to analyze the trade-off between performance and complexity offered by the proposed approach.
> > >
> > > This is a good point and we agree that inclusion of more complex baselines would help observe the performance-complexity tradeoff. As mentioned in our response to reviewer **brRc**, it is a bit challenging to find a suitable baseline domain adaptation method, that can be applied to our problem off-the-shelf. It seems like FADA (Motiian et  al., 2017a) is the closest work that fits this category.
> > >
> > > > Maybe the authors could include a future work section in the conclusion (if they assume to continue research on this topic) and discuss the generality of this approach. What are other possible application areas?
> > >
> > > Please note that we have included a few lines on future work in the conclusion section of the paper.
> > >
> > > > The addressed problem is of interest and the authors test the proposed technique both on simulate and real data. However, the main technical contribution of the paper relies on the well-known property of Gaussian distributions (hence the paper provides only marginal novelty) and the two working hypothesis (same number of components between the two Gaussian distributions and a one-to-one correspondence between the components) are not properly discussed from a practical viewpoint. Finally, considering the fact that the authors promote their approach because it is fast and light-weight, a proper complexity analysis is missing in the paper (the one in the appendix is partial and does not seem to include the cost of the MAP rule).
> > >
> > > We sincerely hope that the changes made to the paper during the revision and our responses here have addressed some of these concerns.

---

### Official Review · Reviewer_wact · 2021-10-29

**Correctness:** 3
**Technical Novelty And Significance:** 2
**Empirical Novelty And Significance:** 2
**Recommendation:** 3
**Confidence:** 3

**Main Review:**

Strengths

The need for domain adaptation in communication systems is practically important and the paper does a good job motivating it. The paper is reasonably well written and provides a good background on applications to wireless communications.

Weaknesses
- On the theory side: the proposed domain adaptation method involves maximizing a regularized log-likelihood and is optimized using known methods such as BFGS quasi-newton methods.
- On the application side:  the block error rates of 0.3 etc in Fig. 5 are too high to be of practical interest in communications applications. I would argue that none of the schemes presented are practical for the models of interest.
- The authors notes that for AWGN to Ricean fading, the performance of the adapted network is sometimes worse than the baseline. The authors’ explanation for this phenomenon is that the “distribution of the two domains are not very different. ”. However, if the distribution of the two domains are close, we should expect the performance of the adapted system stays the same as or slightly better than the baseline. Furthermore it should be noted that the AWGN to Ricean fading is of practical interest as both models involve a dominant line-of-sight component.


**Summary Of The Paper:**

The paper considers an application of deep-learning to communication systems. Due to the nature of deep neural networks that always learn to work on training distribution, the current deep-learning based methods often perform badly when there is a domain shift between training and testing data. And there is no surprise that this problem occurs in communication systems using neural networks. The authors propose  a fast and light-weight method for adapting a Gaussian mixture density network (MDN) using only a small set of target domain samples. This method is well-suited for the communication setting, where the distribution of target data changes rapidly (e.g., a wireless channel), making it challenging to collect a large number of samples and retrain. The authors propose a method for adapting the auto-encoder without modifying the encoder and decoder neural networks, and adapting only the MDN model of the channel. The method utilizes feature transformations at the decoder to compensate for changes in the channel distribution, and effectively present to the decoder samples close to the source distribution. Some experimental results are provided to evaluate the proposed approach.

**Summary Of The Review:**

The paper has significant weaknesses in the proposed applications side and has limited novelty on the algorithmic side. I therefore do not recommend acceptance.

---

> ### Author Response · Authors · 2021-11-22
> **Author response to Reviewer wact**
>
> We thank the reviewer for their comments and feedback on the paper. We are encouraged that the reviewer found the problem to be practically important and well-motivated in our paper. We respond to the individual comments and questions below.
>
> > On the theory side: the proposed domain adaptation method involves maximizing a regularized log-likelihood and is optimized using known methods such as BFGS quasi-newton methods.
>
> Could the reviewer please clarify why this is a weakness of the proposed method? The conditional or posterior log-likelihood is used as the data-dependent term in the adaptation objective (Eqs. 7 and 8). This is a principled choice commonly used for learning/adapting generative models. The conditional log-likelihood can be motivated as  the sample estimate of the KL-divergence  between the true (unknown) conditional density $P(\mathbf{y} | \mathbf{x})$ and the model estimate of the conditional density $P_{\theta}(\mathbf{y} | \mathbf{x})$. The posterior log-likelihood is more suitable when the end-to-end goal is to improve the discriminative performance of the autoencoder.
>
> The quasi-newton BFGS method  is used for optimization because of the small number of parameters and samples involved in the adaptation objective. We also explored using the SGD-based Adam optimizer, and found BFGS to be a better choice.
>
>
> > On the application side: the block error rates of 0.3 etc in Fig. 5 are too high to be of practical interest in communications applications. I would argue that none of the schemes presented are practical for the models of interest.
>
> The block error rates of around 0.3 are observed only when the source and target channels distributions are very different, i.e., AWGN to Uniform fading. A plot of the channel output distribution for an AWGN source channel and Uniform fading target channel, with a 16-QAM constellation and the same SNR of 14dB for both channels, can be found in Figure 8 in Appendix D. This illustrates the significant distribution change even when the SNR of both channels is the same. We report results on such large distribution changes to show the value of adaptation using only a small target sample. Increasing the SNR to values larger than 20dB would likely lead to smaller block error rates. Also, note that the use of error correction coding (ECC) at the bit level can lead to significant reduction in the bit error rates (compared to the block error rates). The use of ECC is orthogonal to the autoencoder-based learning, which primarily focuses on the symbol level encoding and decoding.
>
> From Fig. 5.a  and 5.b, we observe that the block error rates can be much lower (order of 1e-4) when the target channel is Ricean fading. An order of magnitude lower error rates (1e-5 to 1e-6) are observed when the target channel is AWGN. We did not report this result in the paper because it is very simplistic to adapt to an AWGN channel.
>
>
> > The authors notes that for AWGN to Ricean fading, the performance of the adapted network is sometimes worse than the baseline. The authors’ explanation for this phenomenon is that the “distribution of the two domains are not very different. ”. However, if the distribution of the two domains are close, we should expect the performance of the adapted system stays the same as or slightly better than the baseline. Furthermore it should be noted that the AWGN to Ricean fading is of practical interest as both models involve a dominant line-of-sight component.
>
> The reviewer is right in pointing out that when there is minimal or no change in the distribution, the proposed method should maintain a similar block error rate as the original autoencoder. In principle, the KL-divergence based regularization in the adaptation objective should handle this scenario using a large $\lambda$. A possible explanation is that the value of $\lambda$ set automatically using the validation metric (Appendix C.3) is not optimal in some cases.
>
> In practice, the proposed adaptation method should be combined with a front-end detection method that can perform a statistical test to determine if there is a significant change in the distribution that requires adaptation. This would help reduce the frequency with which adaptation is performed, as well as reduce the chance of the adaptation leading to worse performance when there is no distribution change. We leave this aspect, of detecting when there is a distribution change and when adaptation is needed, as future work.

---

> > ### Author Response · Authors · 2021-11-22
> > **Author response to Reviewer wact (continued)**
> >
> > > The paper has significant weaknesses in the proposed applications side and has limited novelty on the algorithmic side.
> >
> > An important focus of the proposed work is to ensure that it is practically applicable to the wireless communication setting. Since end-to-end training of the channel and autoencoder requires expensive data collection and is much slower than adaptation (decreased data throughput and increased latency), it cannot be done frequently. On the other hand, the small number of samples required for adaptation can be easily collected via the packet header (preamble), which is regularly transmitted along with the data. The proposed method is also computationally efficient because of the small number of parameters being optimized, and the fact that the encoder and decoder networks are not modified. While a number of recent papers [O’Shea & Hoydis, 2017; D\”{o}rner et al., 2018; Aoudia & Hoydis, 2019; O’Shea et al., 2019; Ye et al., 2018; Wang et al., 2017] have focused on the problem of end-to-end learning, we are one of the first (to the best of our knowledge) to focus on the domain adaptation aspect of the problem, which can make it more practical to deploy an autoencoder-based communication system under changing channel conditions. We do however agree that there are still aspects of our method (theoretical and computational) that can be improved to make it more performant and efficient.

---

### Official Review · Reviewer_9VpG · 2021-11-01

**Correctness:** 3
**Technical Novelty And Significance:** 3
**Empirical Novelty And Significance:** 2
**Recommendation:** 6
**Confidence:** 4

**Main Review:**

1. The authors only mentioned adapting the MDN model in the abstract and introduction. Readers will be confused why it can be faster than retraining the autoencoder. They revealed that they only train a very small adaptation layer rather than the MDN model on page 3. In my opinion, they should put figure 1 on page 1 to avoid confusion about the speed.

2. When discussing the shortcoming of existing solutions, the author mentioned that the rapidly changing channel leaves not enough time to collect the data to train the autoencoder. But they have the same problem here, they only have a tiny dataset to train the adaptation layer, why overfitting will not happen if they only have such a small dataset?

3. Where does the small labeled dataset come from, and how often to recollect such a dataset and re-run the adaptation?  As the authors mentioned, the channel is changing rapidly, if the channel has changed, the adaptation layer needs to be updated. But collecting the dataset too frequently will leave not enough time to transmit actual data. How to solve this problem?

4. The regularization constant lambda needs to be set carefully for a given pair of source and target distributions. The dependency of lambda choice on the source-target distribution pair makes the proposed solution very impractical. Why not use a "dynamic'' lambda like in ADMM? Is it because a fixed lambda converges faster?

5. No benchmark comparison. There are existing solutions that retrain the entire MDN. The authors should compare with them to show that retraining a small adaption layer does not cause too much accuracy loss compared to retraining the entire MDN.

**Summary Of The Paper:**

This paper proposes an end-to-end learning framework for communication systems using autoencoders. The basic idea is to only retrain a small adaptation layer between the MDN channel model and the decoder. Compared to retraining the autoencoder or the entire MDN channel model, the proposed solution runs faster and fits the dynamic communication channel environment better.

**Summary Of The Review:**

Overall, the proposed idea looks OK. But it is unclear why training with such a small dataset will not lead to overfitting. Also, the overhead of recollecting the labeled dataset and retraining under a rapidly changing channel is not discussed.

---

> ### Author Response · Authors · 2021-11-23
> **Author response to Reviewer 9VpG**
>
> We thank the reviewer for their comments and feedback on the paper. Please find our responses to your questions and comments below.
>
> > 1. The authors only mentioned adapting the MDN model in the abstract and introduction. Readers will be confused why it can be faster than retraining the autoencoder. They revealed that they only train a very small adaptation layer rather than the MDN model on page 3. In my opinion, they should put figure 1 on page 1 to avoid confusion about the speed.
>
> Thank you for pointing out this issue. In the revised paper, we have attempted to clarify the idea of adapting the MDN based on a small parameter-transformation layer (referring to Fig. 1), which makes it faster to adapt compared to retraining the MDN.
>
> > 2. When discussing the shortcoming of existing solutions, the author mentioned that the rapidly changing channel leaves not enough time to collect the data to train the autoencoder. But they have the same problem here, they only have a tiny dataset to train the adaptation layer, why overfitting will not happen if they only have such a small dataset?
>
> The rapidly-changing channel distribution makes it challenging to collect sufficient data for retraining the MDN and autoencoder. This would require transmitting a large number of null data (not actual information) packets for training, which can reduce the effective data rate and increase latency. However, the small number of samples required for adaptation can be easily obtained from the preambles that are transmitted with all data packets. This would not interrupt the transmission of actual data.
>
> The proposed method safeguards against overfitting via the following two mechanisms: *1)* small number of parameters being optimized and *2)* regularization based on KL-divergence in the adaptation objective. The former significantly reduces the search space of possible adapted solutions, and the latter can ensure that the distribution of the adapted MDN is not very different from that of the original MDN. The hyper-parameter $\lambda$ controls the extent of exploration that is allowed during adaptation.
>
> > 3. Where does the small labeled dataset come from, and how often to recollect such a dataset and re-run the adaptation? As the authors mentioned, the channel is changing rapidly, if the channel has changed, the adaptation layer needs to be updated. But collecting the dataset too frequently will leave not enough time to transmit actual data. How to solve this problem?
>
> As mentioned earlier, the small dataset for adaptation can be easily obtained from the preambles that are transmitted with all data packets. This does not interfere with the transmission of actual data, and hence is a practical approach that can be done frequently.
>
> The question of when and how frequently to perform adaptation can be handled by a pre-processing step that detects when there is a significant change in the channel distribution based on statistical testing of the generative model. The pre-processing step can also monitor the performance of the MDN channel and the autoencoder to make this determination. Including such a detection step as part of the adaptation process is practically important, and we leave this as future work.
>
> > 4. The regularization constant lambda needs to be set carefully for a given pair of source and target distributions. The dependency of lambda choice on the source-target distribution pair makes the proposed solution very impractical. Why not use a "dynamic'' lambda like in ADMM? Is it because a fixed lambda converges faster?
>
> The value of $\lambda$ controls the extent of adaptation that is allowed from the source to the target distribution. If one has prior knowledge such as the distribution is not likely to change significantly, then larger values of $\lambda$ can be used. To avoid the hassle of having to set $\lambda$, we propose to use a validation metric -- the (negative) conditional log-likelihood of the inverse-transformed target dataset under the source domain MDN (Appendix C.3). We minimize the adaptation objective for different values of $\lambda$ over a search set, and choose the $\lambda$ that leads to the smallest validation metric. It is not clear why the reviewer thinks that having to set $\lambda$ makes the method impractical. It is not uncommon for domain adaptation methods, e.g., DANN (Ganin et al., 2016), to have tunable hyper-parameters.
>
> Based on our ablation experiments (Appendix D.7) on the choice of $\lambda$, we think there is a range of values (not too small or large) where the method can perform well. Using a dynamic $\lambda$, or decreasing $\lambda$ from a large to a small value according to a schedule are potentially good alternatives to explore. But they may be harder to motivate compared to using a fixed $\lambda$.

---

> > ### Author Response · Authors · 2021-11-23
> > **Author response to Reviewer 9VpG (continued)**
> >
> > > 5. No benchmark comparison. There are existing solutions that retrain the entire MDN. The authors should compare with them to show that retraining a small adaption layer does not cause too much accuracy loss compared to retraining the entire MDN.
> >
> > In the revised paper in Appendix D.3, we have included comparison with a fully-retrained autoencoder and MDN channel (which would be the best-case performance). Since the retraining benchmark uses much more data compared to adaptation, they can achieve much lower block error rates compared to the adaptation methods. However, they are much more data and computationally intensive.
> >
> > > Overall, the proposed idea looks OK. But it is unclear why training with such a small dataset will not lead to overfitting. Also, the overhead of recollecting the labeled dataset and retraining under a rapidly changing channel is not discussed.
> >
> > We hope that our responses have clarified some of the reviewer’s concerns about overfitting and the practicality of collecting data frequently for adaptation.

---

> > > ### Comment · Reviewer_9VpG · 2021-11-30
> > > **The author's response looks good to me**
> > >
> > > The author's response has addressed most of my previous concerns with this paper. I already gave a positive recommendation, so there is no need to change.

---

### Official Review · Reviewer_8ana · 2021-11-01

**Correctness:** 2
**Technical Novelty And Significance:** 2
**Empirical Novelty And Significance:** 2
**Recommendation:** 3
**Confidence:** 4

**Main Review:**

Strength:
(1)	The paper proposes a new MDN based adaptation model for e2e communication system.

(2)	The proposed method is verified by both synthetic and real-world datasets.

Here are some detailed comments:

(1)	Is the adaptation method applicable to other e2e autoencoder models, e.g., GAN or VAE. In another words, is there any specific advantage of selecting MDN?

(2)	Related works on transfer learning miss important references on semi-supervised domain adaptation, e.g., [ref1] and [ref2]. Please also find more semi-supervised transfer methods online.

[ref1] Adaptive Knowledge Transfer based on Transfer Neural Kernel Network

[ref2] Adaptive Consistency Regularization for Semi-Supervised Transfer Learning

(3)	Do you also assume the marginal distribution of x, (or s) is also different between the source and target domains?

(4)	Diagonal covariance implies the inputs are independent, which is a quite strong assumption, it is necessary to work on the general covariance case.

(5)	It is unclear how to ensure $\mathbf{C}_i$ to be diagonal. In fact, I think it is infeasible to make $\mathbf{C}_i$ diagonal according to the derivation in appendix C.1.

(6)	What’s difference between $\hat{\pi_i}$ and $\hat{\alpha_i}$.

(7)	The objective eq.(6) calculates the KL-divergence of two gaussian mixtures. This requires the data size of the source and target to be the same. Is the source data size reduced to match the small number of target data points. In this way, it is not very convincing to estimate the true conditional probability using limited data points.

(8)	The comparison methods lack the state-of-the-art semi-supervised domain adaptation methods, which is less convincing.

(9)	Experiments on simulated and real-world datasets are imbalanced. More real-world experiments are expected.  Moreover, the current setting of section 5.3 is not very clear. How is the transfer setting formulated, i.e., please give the source and target clearly.



**Summary Of The Paper:**

In this paper, the authors propose an MDN based adaptation model to e2e communication system. Specifically, it transforms the inputs to the decoder such that their class-conditional distributions are close to that of the source domain. Experiments are done on both synthetic and real-world datasets.

**Summary Of The Review:**

Overall, the paper needs further improvements to reach ICLR standard. See detailed comments above.

---

> ### Author Response · Authors · 2021-11-21
> **Author response to Reviewer 8ana**
>
> We thank the reviewer for their comments and feedback on the paper. Please find our response to the individual points (1 to 5) under the main review below:
>
> 1. Just to clarify, the choice of a GAN, VAE, or MDN pertains only to the generative model of the channel, and not the end-to-end autoencoder. The proposed method is specific to a mixture density network channel model. Adaptation of other generative models such as GAN and VAE using a small number of samples is an interesting but challenging problem. The choice of MDN as the channel model for this problem comes with the following advantages:
>     * The conditional density corresponding to the MDN is a mixture of Gaussians, whose parametric form is easy to adapt using symbol and component-conditional affine transformations. The number of affine transformation parameters is much smaller than the number of MDN parameters, making the adaptation less susceptible to overfitting on the small dataset from the target distribution.
>     * The MDN choice makes it possible to derive a closed-form KL-divergence between the source and target Gaussian mixtures under some mild assumptions. This is used as a regularization term in the adaptation objective, which allows to directly constrain the change in distribution.
>      * The affine transformation property between the mixture of Gaussians allows us to design an input transformation method at the decoder (Sec 4.1) that can compensate for changes in the distribution of the channel output. This same property also allows us to propose a validation metric that is used for selecting the hyper-parameter lambda (Appendix C.3).
>
>
> 2. Note that our problem setting is different from that of semi-supervised domain adaptation (SSDA). In SSDA, one has access to a large unlabeled dataset and a small labeled dataset from the target domain. In our problem, the target domain only has access to a small labeled dataset and does not have any unlabeled data. We do not consider [ref1], [ref2], and similar works in the semi-supervised or multi-task setting to be closely related to our paper.
>     * The paper [ref1] deals with the problem of transferring regression models between a source and target task that are different but related. In our problem, we do not consider transfer across tasks; rather we consider distribution changes within the same classification task.
>     * The paper [ref2]  deals with the problem of semi-supervised transfer learning between two related tasks. The target domain has access to both labeled and unlabeled data.
>
>
> 3. We assume that the marginal distribution of `x` or `s` does not change significantly between the source and target domains. In other words, the class-prior distribution is assumed to be similar in both the domains. We provide more clarity on the distribution change below:
>     * $s \in \lbrace1, \cdots, m\rbrace$ is the class label and target of the decoder classifier.
>     * $\mathbf{x} = \mathbf{E}_{\theta_e}(\mathbf{1}_s)$  is the encoded vector representation of the class label $s$. The prior distribution of $s$ and $x$ are equal because of the one-to-one mapping of the encoder.
>     * $\mathbf{y}$ is the feature vector (input) of the decoder classifier. The decoder learns the class posterior $P_{\theta_d}(s \text{ }|\text{ } \mathbf{y})$.
>     * We do *not* make the covariate shift assumption. The joint distribution of the feature vector and class label in our problem is $P(\mathbf{y}, s) = P(s \text{ }|\text{ } \mathbf{y}) \text{ }P(\mathbf{y})$. We assume that *both* $P(\mathbf{y})$ and $P(s \text{ }|\text{ } \mathbf{y})$ can change from the source to the target domain.
>
>
> 4. This is not true in our setting. The reviewer is referring to a single multivariate Gaussian, for which diagonal covariance implies independence. For a mixture of Gaussians with more than one component, diagonal covariance does not imply independence. In our setting, the conditional density of the channel $P(\mathbf{y} \text{ }|\text{ } \mathbf{x})$ is a mixture of Gaussians for each $\mathbf{x}$. So this is not a strong assumption, and the proposed method applies to the general covariance setting with minor modifications (e.g. $\mathbf{C}_i$ becomes full instead of diagonal).
>
>
> 5. If the covariance matrix is constrained to be diagonal in both the source and target Gaussian mixtures, then $\mathbf{C}_i$ has to necessarily be diagonal according to the equation $\widehat{\mathbf{\Sigma}}_i = \mathbf{C}_i \mathbf{\Sigma}_i \mathbf{C}_i^T$. Note that $\mathbf{C}_i$ is simply the parametrization chosen by design, and there is no question of infeasibility here.

---

> > ### Author Response · Authors · 2021-11-21
> > **Author response to Reviewer 8ana (continued)**
> >
> > Please find our response to the individual points (6 to 9) under the main review below:
> >
> > 6. $\pi_i \in [0, 1]$ is the mixture prior probability of component $i$, and $\alpha_i \in \mathbb{R}$ is the corresponding un-normalized logit. They are related via the softmax function as $\pi_i = e^{\alpha_i} / \sum_j e^{\alpha_j}$. Adapting the unconstrained logits is easier than adapting the corresponding probabilities. We apologize that this was not clearly defined in the paper. We have now revised the paper with this definition in section 3.1.
> >
> >
> > 7. We believe that the reviewer has misunderstood how the KL-divergence (KLD) is calculated. The KLD in Eq. (6) does not depend on the data sets from the source and target domains. It is *not* a sample estimate of the KLD. Rather the analytical closed-form expression of the KLD, independent of any data sample, is used as derived in Appendix C.2. In Eq. (22), the first term is the expected KLD between the mixture prior probabilities, and the second term is the expected KLD between the component Gaussian densities.
> >
> >
> > 8. Based on our earlier response to comment 2, we do not think semi-supervised domain adaptation methods are appropriate for comparison with our method.
> >
> >
> > 9. This is a two-part response:
> >      * A large part of our experimental evaluation is on simulated channel variations that are commonly used for modeling communication channels. Also, these distribution changes (of varying SNR) can be quite challenging for adaptation in the small-sample setting. To the best of our knowledge, there are no real-world, publicly-available datasets suitable for this domain. Therefore, we have performed real over-the-air data collection and evaluation using a mmWave FPGA testbed (details in Appendix D.1). We plan to conduct more extensive experiments using this setup as future work for a communication-focused paper.
> >     * In section 5.3, we introduced distribution changes via IQ imbalance based distortions to the symbol constellation, and evaluated the adaptation performance as a function of the level of imbalance. The source domain would be the original channel, the over-the-air link between the transmitter and receiver on which the training data is collected. This source domain data is used for training the channel and the autoencoder. The target domain would be a modification of the source domain where the symbols used by the transmitter (e.g., 16-QAM) are distorted by modifying the in-phase and quadrature-phase (IQ) components of the RF signal. This causes a change in the distribution observed by the decoder, leading to a drop in performance without any adaptation. Due to the page limit, we have added details clarifying the above points in Appendix D.1 of the revised paper.
> >
> > We believe that the reviewer’s assessment that “several of the paper’s claims are incorrect or not well-supported” stems from some mis-understanding of our paper. We hope that the above responses help address the concerns. If the reviewer still finds that there are incorrect claims or statements, we kindly request them to provide more details.

---

### Official Review · Reviewer_brRc · 2021-11-02

**Correctness:** 2
**Technical Novelty And Significance:** 2
**Empirical Novelty And Significance:** 2
**Recommendation:** 5
**Confidence:** 3

**Main Review:**

*Strong Points*
1. Their proposed method is simple, but shows solid gain over baselines when target samples are limited. Applying affine transformation to get the parameters of the Gaussian mixture model in the target domain is a convincing approach. And, their simple approach is showing good gains in several cases.
2. Judging from their description of the related work, this problem is not much studied in the wireless communication area. Exploring a new and practical scenario is a strong point in this paper.

*Weak Points*
1. The method is basically specialized for wireless communication with an autoencoder. Although the first several sentences of the abstract seem to describe more general scenarios, their methods are tailored for the wireless communication experiments, and unclear whether the proposed method is widely applicable in other recognition problems, e.g., image classification and sentiment analysis.
2. It is hard to judge the novelty of the proposed method since their baseline is very naive. There is no baseline borrowed from the literature of few-shot classification or domain adaptation work for limited target samples, e.g., "Few-Shot Adversarial Domain Adaptation, Neurips 2017". I understand that it may be hard to directly borrow such baselines since they are not designed for wireless communication.
3. Related to 2, they are not covering several few-shot domain adaptation works, such as  "Few-Shot Adversarial Domain Adaptation, Neurips 2017". Their work is not the first to study the problem. It was confusing.


**Summary Of The Paper:**

This paper presents a method for domain adaptation, where we have access to only a small number of labeled target samples and aim to adapt a model with a small training cost. They propose a fast and light-weight method for adapting a Gaussian mixture density network
(MDN). Specifically, they transform the parameters of source Gaussian mixture models to those of the target domain by applying affine transformations.  They make the assumption of one-to-one pairing between the components of the mixture models. To optimize a model, they regularize training by the Kullback-Leibler divergence between two paired mixture models. This method is specialized for e2e learning for a communication system using an auto-encoder.  They evaluate the performance of the model on several adaptation scenarios in wireless communication and show a good advantage over baselines when enough target samples are not available.

**Summary Of The Review:**

From the weakness I described above, I recommend this paper below the acceptance threshold. The work may have high importance in the application to wireless communication, but lacks discussions related to existing domain adaptation work and comparison to them.

---

> ### Author Response · Authors · 2021-11-23
> **Author response to Reviewer brRc**
>
> We thank the reviewer for their comments and feedback on the paper. We are encouraged that the reviewer found the proposed work to be convincing and practically important. We respond to the individual comments and questions below.
>
> > The method is basically specialized for wireless communication with an autoencoder. Although the first several sentences of the abstract seem to describe more general scenarios, their methods are tailored for the wireless communication experiments, and unclear whether the proposed method is widely applicable in other recognition problems, e.g., image classification and sentiment analysis.
>
> We have now revised the abstract so as to not overstate the generality of the proposed work to other application domains. The reviewer is correct in pointing out that the proposed domain adaptation is specific to the autoencoder based end-to-end communication problem. Since the decoder is a general DNN classifier, the method proposed for adaptation at the decoder side (section 4) can be applied to other domains that are not high-dimensional. We also acknowledge this under “Limitations and future work” of the Conclusions section.
>
> >  It is hard to judge the novelty of the proposed method since their baseline is very naive. There is no baseline borrowed from the literature of few-shot classification or domain adaptation work for limited target samples, e.g., "Few-Shot Adversarial Domain Adaptation, Neurips 2017". I understand that it may be hard to directly borrow such baselines since they are not designed for wireless communication.
>
> In the “Related work” section of the revised paper, we have cited and discussed the paper on few-shot adversarial domain adaptation (FADA). We thank the reviewer for pointing out this closely-related paper.
>
> Both our paper and FADA deal with a small labeled dataset from the target domain and no unlabeled data (supervised domain adaptation). A key distinction between our problem setting and that of FADA is that we are interested in the test-time domain adaptation of an existing classifier from the source domain. However, FADA focuses on training-time domain adaptation, i.e. a classifier is trained from the combined source and target datasets using the novel idea of adversarially training a joint domain-class discriminator (DCD). In our problem setting, we expect the adaptation method to be able to quickly adapt using new batches of data from a changing target distribution. A challenge with adopting FADA for this problem is that it would require to be retrained for each new target data batch, which can be computationally challenging. We now discuss this in the related work section of the paper.
>
> Another relatively minor challenge with applying a method like FADA to this problem is that it would not be able to adapt the generative model of the channel (MDN). It can adapt the decoder classifier, but this would not involve the channel model at all. In our problem, it is important to adapt both the channel model and the decoder, to keep track of the changing channel conditions.
>
> > Related to 2, they are not covering several few-shot domain adaptation works, such as "Few-Shot Adversarial Domain Adaptation, Neurips 2017". Their work is not the first to study the problem. It was confusing.
>
> We hope the revised related work section addresses this concern.

---

### Author Response · Authors · 2021-11-24
**Summary of changes to the revised paper**

We thank all the reviewers for their time and valuable feedback on our paper. We have revised the paper based on the feedback and attempted to incorporate most of the suggestions from the reviewers. The key changes in the revised paper are **highlighted in blue**, and we list them below.

* We have revised the abstract to better reflect the main focus and contributions of the paper.

* We have updated the related work section to add two important references suggested by Reviewer brRc. We contrast our work with this related paper (Motiian et al. (2017a)), and explain the challenge of applying their approach to our problem setting.

* We have added some details to section 3.1 to explain the problem setup and notations better.

* We now clearly state the assumptions about the source and target Gaussian mixtures and provide an explanation for each (under “Assumptions and Key Insight”).

* In section 5.2, we have revised the “observations and takeaways” a little and added a forward reference to Appendix D.3 and D.4 which provide additional results.

* In section 5.3, we have added a forward reference to Appendix D.2, which provides more details on the setup for the real FPGA experiment.

* We have revised the computational complexity analysis in Appendix C.4 to include more details about the individual decoder adaptation methods.

* We have added Appendix D.3, which includes comparison of the autoencoder adaptation methods with a benchmark retrained autoencoder. These results are given in figures 10 and 11.

* We have added Appendix D.4, which compares the two variants of the decoder adaptation methods we proposed in section 4. These results are given in figures 12, 13, and 14.

* Figure 8  was added to show a distribution plot of the simulated channels.

* Figure 9 was added to show plots of the constellation learned by the autoencoder.

---

### Decision · Program_Chairs · 2022-01-20

**Decision:**

Reject

**Comment:**

The paper proposes a domain adaptation method that is specific to auto-encoder and wireless domain. The proposed method shows solid gain over baseline and is simple. There were multiple complaints on reviewer's side regarding clarity of the abstract and application of the work and related work that was responded to by the authors during the rebuttal period. However, the modifications were not enough to address all concerns. Mainly, the assumptions for the method to work such as source and target having the same number of components and diagonal covariance. It would help the paper to discuss the cases where the model fails. In addition, the paper will benefit from a stronger baseline.